# Induced organoids derived from patients with ulcerative colitis recapitulate colitic reactivity

Samaneh K. Sarvestani[1], Steven Signs[1], Bo Hu[2], Yunku Yeu[2], Hao Feng [3], Ying Ni[2], David R. Hill [4], Robert C. Fisher[1], Sylvain Ferrandon[1], Reece K. DeHaan [5], Jennifer Stiene[1], Michael Cruise[6], Tae Hyun Hwang [2], Xiling Shen [7], Jason R. Spence [4,8,9] & Emina H. Huang [1,5 ✉]

The pathogenesis of ulcerative colitis (UC), a major type of inflammatory bowel disease, remains unknown. No model exists that adequately recapitulates the complexity of clinical UC. Here, we take advantage of induced pluripotent stem cells (iPSCs) to develop an induced human UC-derived organoid (iHUCO) model and compared it with the induced human normal organoid model (iHNO). Notably, iHUCOs recapitulated histological and functional features of primary colitic tissues, including the absence of acidic mucus secretion and aberrant adherens junctions in the epithelial barrier both in vitro and in vivo. We demonstrate that the CXCL8/CXCR1 axis was overexpressed in iHUCO but not in iHNO. As proof-of-principle, we show that inhibition of CXCL8 receptor by the small-molecule non-competitive inhibitor repertaxin attenuated the progression of UC phenotypes in vitro and in vivo. This patient-derived organoid model, containing both epithelial and stromal compartments, will generate new insights into the underlying pathogenesis of UC while offering opportunities to tailor interventions to the individual patient.

[1] Department of Cancer Biology, Cleveland Clinic Lerner Research Institute, Cleveland, OH 44195, USA. [2] Department of Quantitative Health Sciences, Cleveland Clinic Lerner Research Institute, Cleveland, OH 44195, USA. [3] Department of Population and Quantitative Health Sciences, Case Western Reserve University, Cleveland, OH 44106, USA. [4] Department of Internal Medicine, Gastroenterology, University of Michigan, Ann Arbor, MI 48109, USA. [5] Department of Colorectal Surgery, Cleveland Clinic, Cleveland, OH 44195, USA. [6] Department of Pathology, Cleveland Clinic, Cleveland, OH 44195, USA. [7] Department of Biomedical Engineering, Duke University, Durham, NC 27708, USA. [8] Department of Cell and Developmental Biology, University of Michigan, Ann Arbor, MI 48109, USA. [9] Department of Biomedical Engineering, University of Michigan, Ann Arbor, MI 48109, USA. ✉email: Huange2@ccf.org

Ulcerative colitis (UC) is one of the two principal types of inflammatory bowel disease (IBD) and is a debilitating inflammatory condition of the colon that usually begins in young adulthood[1]. Although the precise pathogenesis is unknown, the lack of mucus secretion, elevated concentrations of soluble inflammatory mediators secreted by the stroma, wounding, and epithelial barrier permeability are thought to play essential early roles in the development and progression of UC[2–5]. Clinical UC is heterogeneous, ranging from acute to chronic inflammation of the colon. As the pathogenesis is unknown, medical interventions are nonspecific, resulting in inconsistent treatment responses[6,7]. Patients with poor response to these therapies develop severe symptoms, ultimately leading to resection of the colon and rectum[8,9]. Because patient-based predictive models, which capture individual heterogeneity, are lacking, a more complete experimental model is essential.

Pluripotent stem cell (PSC)-derived human intestinal organoids have been developed for the normal mid/hind gut[10,11] and colon[12,13] and have distinct advantages in advancing precision medicine[14]. These organoids can be derived from the individual patient and exhibit recapitulation of physiological responses both in vitro and in vivo[10,12,15]. Besides the unique characteristics of PSCs, induced PSC-derived (iPSC) organoids retain the phenotype and genetic background of the individual donor as reported by recent studies modeling both normal and pathological organs[16–18].

In this study, we provide evidence linking recent advances in stem cell biology[19] with intestinal development[10–12,15,20–22] to reprogram colonic fibroblasts isolated from UC patients to iPSCs, followed by directed differentiation to induced human UC organoids (iHUCOs). Notably, iHUCOs have both epithelial and stromal compartments, preserving the colitic phenotype of their tissue of origin, such as responding to wounding by aberrant proliferation, a lack of goblet cells, and having compromised cellular tight junctions in their epithelial barrier.

High-resolution transcriptome-wide analysis of iHUCOs and UC parental fibroblasts revealed pathways and inflammatory mediators in common between both groups compared with their controls, induced human normal organoids (iHNOs) and normal fibroblasts. Specifically, the roles of the inflammatory chemokine CXCL8 and chemotactic cytokine CXCL1 were highlighted in the transcriptome and mesenchymal secretome of iHUCOs and their parental fibroblasts.

As proof-of-principle, we studied the iHUCO functional response to the CXCL8 receptor antagonist, repertaxin, and demonstrated the attenuation of colitic features in iHUCOs both in vitro and in a xenograft model in vivo. Mice have no homolog for human CXCL8. Therefore, this axis is not present in genetic or chemically induced murine models, emphasizing the essential utility of patient-derived UC models.

## Results

**Patterning of induced organoids recapitulates in vitro their primary tissues.** Fresh surgical specimens from colitic and healthy colons were obtained (Fig. 1a), and fibroblasts were isolated[23] (Fig. 1b). We reprogrammed UC fibroblasts to iPSCs; UC fibroblasts were isolated from 6 patients with established chronic colitis along with samples from 4 participants with normal colonic fibroblasts and one from a commercially available source (Supplementary Table 1, Supplementary Fig. 1a, 1b). Pluripotency of the generated iPSCs was confirmed (Supplementary Fig. 1c–l). Next, we applied an established protocol for intestinal organoid generation[11] to direct differentiation of the iPSCs into definitive endoderm (DE), validated by SOX17 and FOXA2 protein expression (Fig. 1a, d), followed by intestinal spheroid formation

(SPH, validated by CDX2 expression, Fig. 1a, e). Normal and UC spheroids were then cultured in Matrigel for 21 days to develop induced human normal organoids (iHNOs), and iHUCOs, respectively. Representative immunofluorescent (IF) staining of iHUCOs including epithelium (stained for CK19) and mesenchyme (stained for VIM) is shown in Fig. 1f.

Both iHNO and iHUCO were characterized by comparison to their matched primary tissues. Hematoxylin and eosin (H&E) staining illustrated distinct epithelial and mesenchymal domains with an interior lumen in organoids (Fig. 1g1, g2). In addition, iHNOs frequently had a simple columnar epithelium, which is representative of the healthy colonic mucosa (Fig. 1g1) whereas iHUCOs had disorganized (pseudostratified and stratified) epithelium (Fig. 1g2). These organoid data appear to recapitulate the presence of greater pseudostratification in colonic mucosa of patients with active UC[24] caused by reparative changes in response to epithelial wounding (Fig. 1g3, g4). The percentages of simple, pseudostratified, and stratified epithelium for normal and UC organoids (Fig. 1h) and their primary tissues (Supplementary Fig. 2a) confirmed a significantly higher ratio of non-columnar epithelium in iHUCOs and colitic primary tissues than in the normal condition.

Immunohistochemical (IHC) staining for the proliferation marker KI67 revealed a more uniform pattern of cellular proliferation throughout the columnar epithelium of iHNOs, similar to the normal primary tissues (Fig. 1i1, i3). In contrast, regions of stratified epithelium in iHUCOs and colitic tissues had extensive and non-uniform epithelial proliferation with greater distribution (Fig. 1i2, i4, j). Epithelial expression of KI67 approached 80% in iHUCO and UC tissues, whereas it was significantly lower in normal tissue. Thus, KI67 expression in the iHUCOs suggests an accelerated rate of epithelial turnover that may be similar to the rate of epithelial cell turnover in colonic mucosa undergoing regeneration in patients with active UC[25].

The intestinal mucus layer secreted by goblet cells in healthy mucosa includes both acid and neutral mucins to protect the epithelial barrier from luminal bacterial penetration[26]. Alcian blue and periodic acid-schiff (AB-PAS) staining revealed the secretion of both types of mucins in the iHNO lumen along with a limited numbers of goblet cells (Fig. 1k1). In contrast, iHUCOs had no or only neutral mucin staining along with a significantly lower number of goblet cells than iHNOs, suggesting a lack of acidic mucin secretory function (Fig. 1k2). The number of goblet cells in UC primary tissues (Fig. 1k4) was significantly lower than in normal tissues (Fig. 1k3, 1l, $P < 0.0001$). These organoid data are similar to the depletion of goblet cells and the mucus layer observed in the colonic mucosa of patients with UC[27]. Impaired function of the intestinal barrier in patients with UC leads to recurrent severe inflammation and eventual wounding. In response to tissue damage, the epithelial compartment switches to repairing the epithelium as demonstrated by an elevated number of proliferative cells (Ki67 positive) and fewer goblet cells than in homeostatic tissue[28].

The expression of CDX2 plays a crucial role in intestinal development, including balancing proliferation with differentiation and epithelial barrier formation[11,29,30]. As expected, uniform and strong expression of CDX2 that was restricted to the epithelium was observed in iHNOs (Supplementary Fig. 2e1) and normal colon tissues (Supplementary Fig. 2e3). In contrast, CDX2 expression was diminished in primary UC tissues and their corresponding organoids ($p < 0.0001$, supplementary Fig. 2e2, e4, and f). Thus, the CDX2 expression patterns in our UC organoids were similar to inflammation-related decrease in CDX2 reported for patients with active UC[31,32].

Recently, SATB2 has been identified as a definitive marker of distal small intestine (ileum) and colonic epithelium in humans[12].

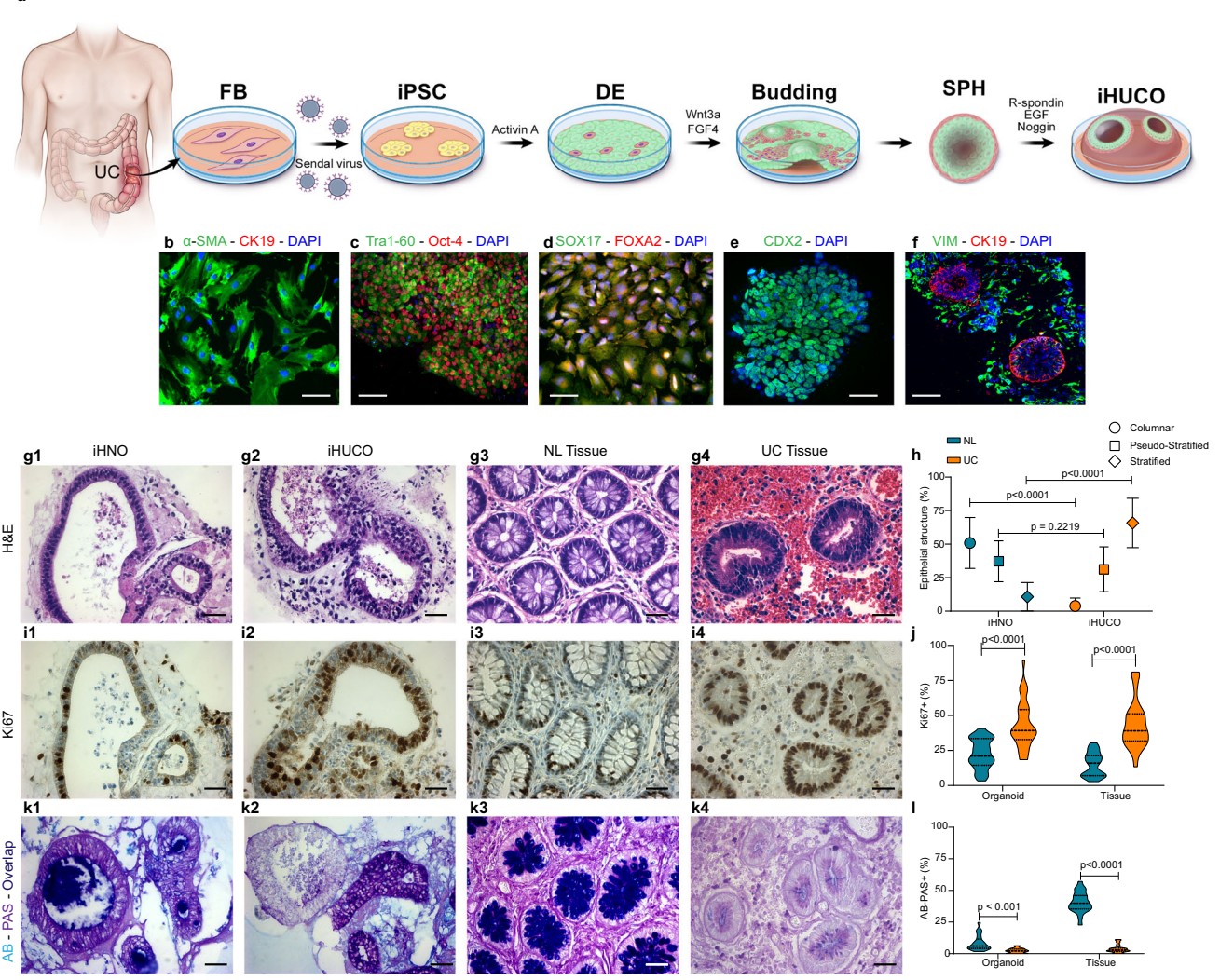

**Fig. 1 Patterning of induced organoids recapitulates in vitro their primary tissues. a** Schematic representation of induced human ulcerative colitis organoid (iHUCO) generation protocol followed by immunofluorescence (IF) staining of key proteins at each stage of development. **b** Fibroblasts (FB) express α-SMA (green) and lack CK19 expression (red). **c** Induced pluripotent stem cells (iPSCs) express Tra-1-60 (green) and Oct-4 (red). **d** Definitive endoderm (DE) expresses SOX17 (green) and FOXA2 (red). **e** Spheroids (SPH) express CDX2 (green), confirming the intestinal identity. **f** iHUCO expresses CK19 (red) in epithelium and vimentin (green) in the mesenchyme. Nuclei in all IF images are counterstained with DAPI (blue). **g1–g4** Representative H&E-stained induced human normal organoid (iHNO) with simple columnar epithelium and iHUCO with stratified epithelium corresponding to the primary tissues. **h** Summarized percentages for epithelial structure—columnar, pseudostratified, or stratified—in iHNO and iHUCO. The median value is indicated in the center of the box plot. The whiskers above and below the plot represent 1 standard deviation (SD) above and below the mean of the data, respectively. For the columnar and stratified epithelial, $P < 0.0001$. For the pseudo-stratified epithelium, $P = 0.22$. **i1–i4** Representative immunohistochemistry (IHC) for Ki67 in iHNO, iHUCO, and their primary tissues. **j** Percentage distribution of the cells positive for Ki67 in the epithelium of organoids and primary tissues, shown as a violin plot. $P < 0.001$ for both comparisons. **k1–k4** Representative AB-PAS staining in iHNO, iHUCO, and their matched primary tissues. **l** A violin plot of the percentage of goblet cells in the organoids and their primary tissues, contrasting NL and UC. $P < 0.001$ for the organoid comparison, and $P < 0.0001$ for the normal tissues. An unpaired, non-parametric, two-sided Mann–Whitney $U$ test was used for all statistical comparisons. IF scale bar = 25 μm, IHC scale bar = 40 μm. $n = 5$ iHNOs (blue) and $n = 6$ iHUCOs (orange). Source data are provided as a Source Data file.

Similar to CDX2, IHC staining revealed significantly lower expression of SATB2 in UC organoids ($p = 0.001$) and primary tissues ($p < 0.0001$) than in normal controls (Supplementary Fig. 2g, h).

Thus, we conclude that iHUCOs phenocopy the features of their primary tissues, including a stratified epithelium, an aberrant rate of epithelial proliferation, and the absence of acid mucin secretion.

**iHUCOs demonstrate aberrant adherens junction formation in the epithelium.** In health, epithelial cells form a physical barrier within the gut lumen that protects the intestine from bacterial and inflammatory cell infiltration[33]. A dynamic combination of different apical intercellular junctions, including tight and adherens junctions, between the epithelial cells maintains this homeostasis[33,34]. Under chronic inflammatory conditions such as UC, the balance in intercellular junctions is disrupted and the integrity of the epithelial barrier is compromised[35,36]. This disruption results in an increase in the paracellular and intracellular space, bacterial invasion, dysregulation of the immune response, and ultimately a damaged, leaky epithelial barrier[2,37,38].

The tight association between E-cadherin/β-catenin plays a main role in intestinal development and maintaining the epithelial integrity[35]. Although accumulation of β-catenin in the

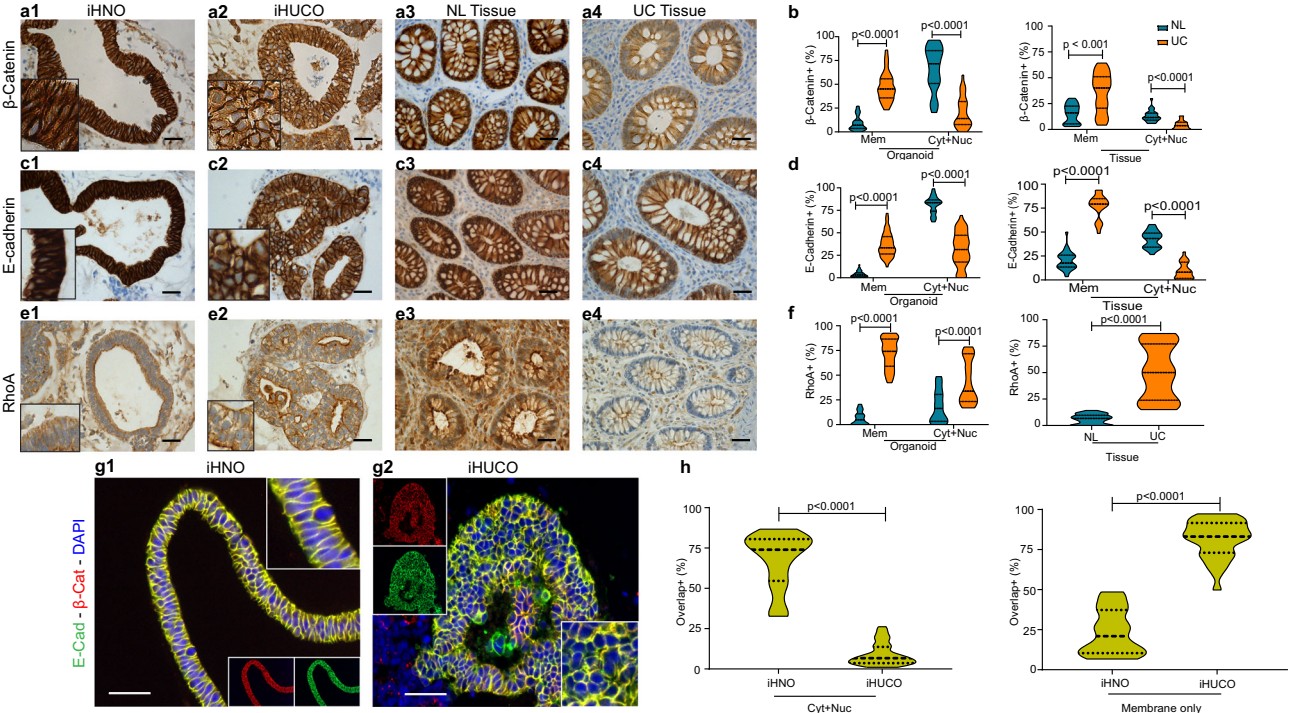

**Fig. 2 iHUCOs demonstrate aberrant adherens junction formation in the epithelium. a, c** Representative IHC for β-catenin and E-cadherin, demonstrating the difference in cellular localization between iHNOs and iHUCOs, as well as their matched primary tissues. **b, d** Violin plots of the percentages of cells expressing β-catenin and E-cadherin in the organoids and primary tissues as distinguished by the cellular compartment: plasma membrane only (Mem) or membrane extended to cytoplasm and nucleus (Cyt + Nuc). **e** Representative IHC for RhoA, demonstrating increased cytoplasmic and membrane expression of RhoA in iHUCO vs. iHNO, and UC vs. normal primary tissues. **f** Violin plot indicating the percentage of cells positive for RhoA according to cellular localization in the organoids, plasma membrane only (Mem), or cytoplasm (Cyt), and the primary tissues. **g** Representative dual IF staining for E-cadherin (green) and β-catenin (red) in iHNO and iHUCO. **h** Violin plots of cells in iHNOs and iHUCOs co-expressing E-cadherin and β-catenin proteins extended to cytoplasm and nucleus (Cyt + Nuc) or in plasma membrane only (Mem). All nuclei are stained with DAPI (blue). An unpaired, non-parametric, two-sided Mann–Whitney $U$ test was used for all comparisons. IF scale bar = 25 μm, IHC scale bar = 40 μm. $n = 5$ iHNOs (blue) and $n = 6$ iHUCOs (orange). Source data are provided as a Source Data file.

cytoplasm and its eventual translocation into the nucleus is essential for canonical Wnt pathway activation and subsequent expression of tight junction proteins, the ability of membrane-associated β-catenin to bind and to co-localize with E-cadherin is a hallmark of adherens junction regulation[39]. We performed IHC to investigate the subcellular localization of β-catenin in the organoids and their matched primary tissues (Fig. 2a). IHNOs demonstrated strong membrane, cytoplasmic, and nuclear expression of β-catenin, suggesting a high degree of protein stability in these organoids (Fig. 2a1). On the other hand, the expression of β-catenin in iHUCOs was mainly limited to the plasma membrane (Fig. 2a2), and the quantification for exclusive expression on the plasma membrane was significantly higher in iHUCOs than iHNOs ($p < 0.0001$). However, the combined cytoplasmic and nuclear β-catenin expression was greater in iHNOs than iHUCOs (Fig. 2b). β-catenin expression was consistently and significantly lower in UC than normal primary tissues (Fig. 2a3, a4, b).

E-cadherin, the main component of the adherens junction complex, had a similar expression pattern as β-catenin in both organoids and primary tissues (Fig. 2c, d). Although E-cadherin was strongly expressed in the plasma membrane, cytoplasm, and nucleus of iHNOs, cytoplasmic and nuclear expression was significantly lower in iHUCOs ($P < 0.0001$, Fig. 2c1, c2, d). Similarly, E-cadherin expression was greater in normal than UC primary tissues (Fig. 2c3, c4, d).

RhoA is one of the dominant regulators of the adherens junction complex, playing roles in cell adhesion and cytoskeleton

organization[40]. To be activated, cytoplasmic RhoA (inactive) first must translocate to the plasma membrane, where it then can regulate downstream functions, including the formation of actin stress fibers (F-actin), focal adhesions, and cell motility[40]. IHC staining for RhoA showed the expression was significantly greater in both cytoplasm and plasma membrane of iHUCOs than in iHNOs, and also in UC than in normal primary tissues (Fig. 2e1–e4, f). These data highlight the propensity for an adherens junction signature in iHUCOs.

To confirm the tight association of E-cadherin/β-catenin proteins, as well as significant differences in their sub-cellular co-localization, we performed dual IF staining in iHNOs and iHUCOs (Fig. 2g). As expected, co-localization in iHNOs extended to the cytoplasm and nucleus, indicating higher stability of both proteins (Fig. 2h) whereas in iHUCOs, co-localization was mostly limited to the plasma membrane, suggesting a propensity towards adherens junctions rather than tight junctions.

To functionally confirm these results, we used the recently described microinjection technique[20] to measure and compare the epithelial barrier permeability for both iHUCOs and iHNOs in real-time. Briefly, we microinjected the organoids with fluorescently labeled 4-kD dextran and imaged them with an inverted microscope fitted with epifluorescent filters for a total of 15 h. Real-time measurement of the barrier permeability showed significantly less dye retention in the iHUCO lumen (~50% of real-time measurement timepoints) than the lumen of iHNOs (Supplementary Fig. 2b). Moreover, CLDN1, a major constituent of the tight junction complexes responsible for normal barrier

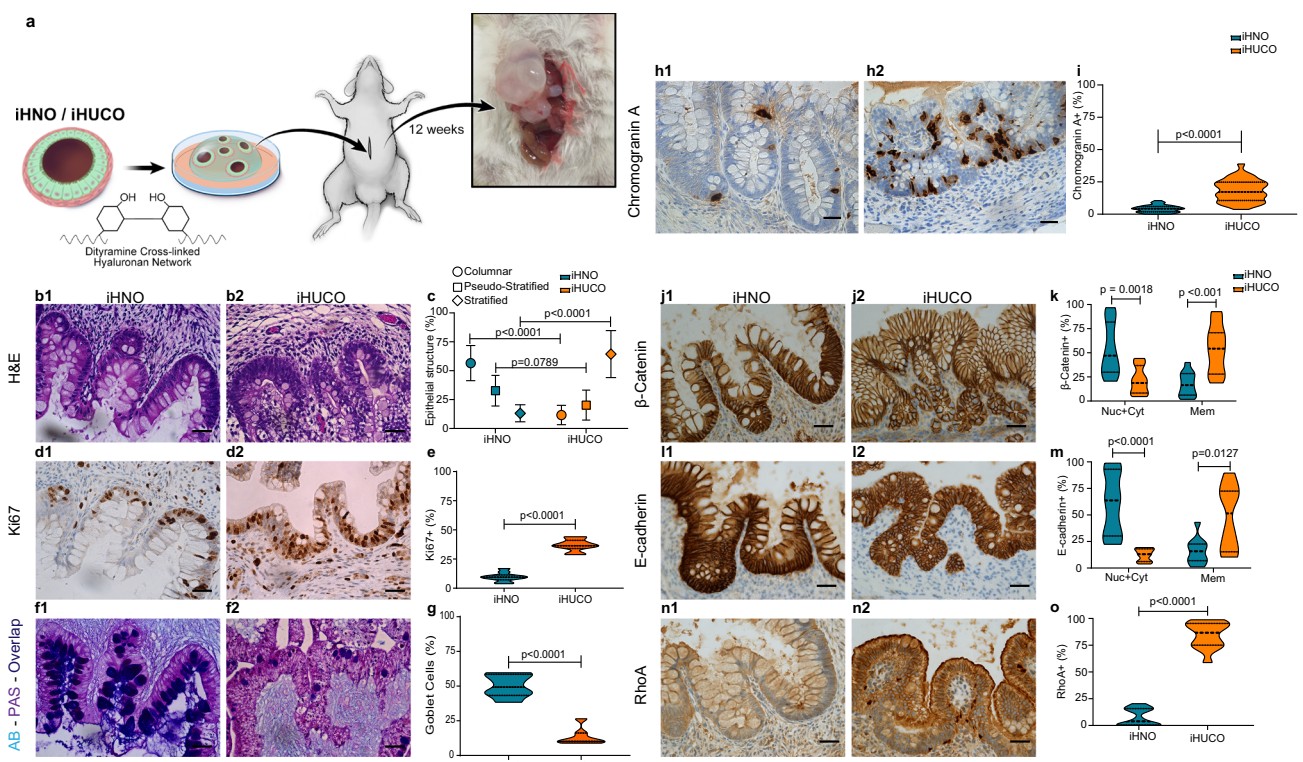

**Fig. 3 iHUCOs retain the colitic phenotype upon omental transplantation. a** Schematic representation of the omental transplantation study. iHN and iHUC organoids were encapsulated in a TS-HA hydrogel (see "Methods" section), and each bead was transplanted into the omentum of NSG mice for 90 days. **b** Representative H&E-stained beads collected at 90 days, confirming the formation of colon-like crypts in iHN and iHUC organoids while retaining the phenotypes: simple columnar epithelia in normal crypts and disorganized epithelium in UC. **c** Summarized percentages for types of epithelial structure in normal and UC epithelium. The median value is indicated in the center of the box plot. The whiskers above and below the plot represent 1-standard deviation (SD) above and below the mean of the data, respectively. **d** Representative IHC for Ki67 in normal and UC organoid-derived crypts confirming the greater rate of proliferation in UC. **e** Percentage distribution of cells expressing Ki67 in normal and UC epithelium. **f, g** Representative AB-PAS staining in normal and UC organoid-derived crypts, highlighting a lack of acid mucin and significantly lower number of goblet cells in UC. **h** Representative IHC for chromogranin A in iHNO and iHUCO colon-like epithelium, indicating a significantly higher number of EECs in UC. **i** Violin plot of cell percentages positive for chromogranin A. **j, l** Representative IHC for β-catenin and E-cadherin showing greater protein expression in normal vs. UC colon-like epithelium. **k, m** Violin plots of cells positive for β-catenin and E-cadherin according to cellular compartment: membrane extended to cytoplasm and nucleus (Cyt + Nuc) or plasma membrane only (Mem). **n, o** Representative IHC for RhoA revealing higher expression in UC vs. normal organoid-derived colon, with quantification of the percentage of cells positive for RhoA. Scale bar, 40 μm. $n = 3$ each. For all comparisons, the unpaired, non-parametric two-sided Mann–Whitney $U$ test was used. Source data are provided as a Source Data file.

function in the epithelium[41] was lower in the epithelium of iHUCOs than in iHNOs (Supplementary Fig. 2c, d).

In summary, our in vitro findings demonstrate similar expression patterns for β-catenin and E-cadherin in the organoids and their primary tissues. We confirmed the co-expression and tight association between β-catenin and E-cadherin in organoids; both proteins co-localized predominantly in the iHUCO plasma membrane whereas expression extended to the cytoplasm and nucleus in iHNOs. Moreover, our functional study of epithelial barrier permeability in organoids with IHC for the tight junction protein CLDN1 confirmed the tight junction compromise in the iHUCO epithelium.

**iHUCOs retain the colitic phenotype upon omental transplantation**. To determine whether iHUCOs retain the colitic phenotype in vivo, we combined the recently reported transplantation protocol for PSC-derived organoids[42,43] using biocompatible TS-HA hydrogel to embed and transplant the organoids into the omentum of host NSG mice (Fig. 3a). At 90 days after transplantation, we collected the organoids. Further characterization by H&E demonstrated that the iHUCOs had a colon-like appearance with disorganized epithelium and short

crypts whereas iHNOs recapitulated simple columnar epithelium (Fig. 3b). Summarized percentages confirmed a significantly higher ratio of stratified epithelium in UC than normal crypts (Fig. 3c). The colitic epithelium also revealed greater proliferation (Fig. 3d1, d2, 3e), a lack of acid mucin secretion, and significantly fewer goblet cells compared with normal epithelium (Fig. 3f1, f2, $p < 0.0001$). As expected, in normal tissue, proliferation was limited to the crypt base (Fig. 3d1), and the secreted mucus included both neutral and acid mucins (Fig. 3f1). The colonic epithelium was also characterized for the expression of CDX2 (Supplementary Fig. 2i) and SATB2 (Supplementary Fig. 2k) proteins. Similar to the in vitro pattern, expression of both proteins was lower in UC than normal crypts (Supplementary Fig. 2j, l).

Enteroendocrine cells, a specialized secretory cell population, in the colonic mucosa act as the functional sensors for any homeostatic dysregulation in the luminal content[44,45]. In response to alterations in the homeostasis of colonic mucosa, such as pro-inflammatory cytokines in the mucosa of the patients with active UC, greater numbers of enteroendocrine cells have been reported in several studies[46–48]. Among the secretory granules of enteroendocrine cells in the gut, chromogranin A, a precursor to several functional peptides, is the most abundant[46].

IHC staining for chromogranin A showed a significantly ($p < 0.0001$) higher number and greater distribution of enteroendocrine cells in the epithelium of UC than normal transplanted organoids, confirming a dysregulation in iHUCO development (Fig. 3h, i). We also confirmed that the combined nuclear and cytoplasmic expression of β-catenin and E-cadherin was lower in UC than normal epithelium (Fig. 3j–m) whereas RhoA expression was significantly higher than in normal colon (Fig. 3n, o; $p < 0.0001$). These expression patterns are consistent with the patterns seen in UC and normal primary tissues.

Both the in vitro and in vivo data confirmed dysregulation in the developmental process of iHUCOs. We observed clinically proven colitic features including changes in crypt histology, aberrant proliferation, more enteroendocrine cells, and adherens vs. tight junction signature in UC epithelium.

**Bulk RNA-sequencing reveals UC signatures in iHUCOs**. To compare the transcriptional activity during differentiation and the development of organoids derived from healthy or colitic iPSCs, we conducted bulk RNA-seq on normal and UC iPSCs, definitive endoderm (DE), spheroids (SPH), and organoids (GSE117345, $n = 3$ each). Principal component analysis (PCA) revealed major transcriptional variations among all genes, driven by the developmental stage (Fig. 4a). To improve our understanding of the similarities and differences between UC and normal intestinal development, we conducted a PCA among DE, spheroid, and organoid stages (Supplementary Fig. 3a). DE, as the first stage of the intestinal development, formed a distinct cluster causing a shift in the gene expression pattern between the UC and normal samples. In contrast, subsequent progression in the development to spheroids and organoids localized the normal and UC closer to each other.

Unsupervised hierarchical clustering of the global gene expression data based on the Spearman rank correlation was performed (Fig. 4b, left). Similar to PCA results, the groups segregated based on the developmental stage rather than disease status, and the organoids formed a discrete cluster separated from DEs and iPSCs but segregated closely with spheroids. Because the developmental stage was the main driver in gene expression, we compared the differentially expressed genes (DEGs) in the sequential stages of development between UC and normal. A Venn diagram of these DEGs (Fig. 4b, right) showed the greatest number of unique genes in the progression from DEs to spheroids.

To confirm the colonic identity of normal and UC organoids at the transcriptome level, we used the list of genes reported recently by Múnera et al comparing the colonic signature with small intestine signature in organoids, as well as primary tissues[12]. Heatmaps of upregulated genes in the colon were generated for the organoids at all three stages of intestinal development (Supplementary Fig. 3c). Although both normal and UC organoids had a $\log_2$ (RPKM) ≥ 1 for the majority of these genes, the expression pattern differed between normal and UC development (Supplementary Fig. 3c1, c2). We listed the top 50 expressed genes in UC and normal organoids as curated heatmaps to highlight this difference (Supplementary Fig. 3e1, e2). A Venn diagram highlights the genes exclusive to normal and UC organoids (Supplementary Fig. 3d).

To explore the molecular states in the spheroid-to-iHUCO progression, we conducted Gene Set Enrichment Analysis (GSEA) and determined the enriched terms by applying complex network analysis[49] using Cytoscape (Fig. 4c). Significant functional terms ($p < 0.01$), including inflammatory and immune response, wound healing, and response to bacteria were identified in iHUCOs, which supports their UC phenotype. We generated curated heatmaps for the key genes belonging to the "inflammatory response" (Fig. 4d, top) and "wound healing" terms (Fig. 4d, bottom) and showed that not only did iHUCOs cluster together as compared with spheroids, but they also showed a significant increase in transcriptional expression of these genes, suggesting a prominent colitic signature.

To identify dominant biological processes enriched in iHUCOs, we applied a ranked list of DEGs to Gene Ontology enrichment analysis (Gorilla, Supplementary Data File 1) and visualized the enriched GO terms using Reduce and Visualize Gene Ontology (REVIGO, Supplementary Data File 1)[50]. The terms clustered semantically along the $X$-axis based on the similarity: highly significant terms ($p < 0.001$) including "actin cytoskeleton organization" and "fiber polymerization," clustered on the left, progressed to "response to mechanical stimuli" and terminated on the right with signaling pathways, including G-protein coupled receptors (GPCR), regulation of interleukin-8 (CXCL8) production, and Rho protein signal transduction (Fig. 4e, Supplementary Fig. 3b). The enrichment scores along with $p$-values for selected GO terms such as TNF, IFN-γ, CXCL1, and CXCL8 signaling are shown in Fig. 4f. Excessive production of IFN-γ and TNF-α inflammatory cytokines is associated with compromised epithelial barrier integrity[51]. TNF-α is also indirectly responsible for activating GPCRs in the epithelium and wound healing. Birkl et al. recently showed that TNF-α promotes intestinal mucosal wound repair in human colonic organoids[52]. Therefore, our data in Fig. 4f showing strong enrichment of similar GO terms confirm the inflamed nature of iHUCOs.

CXCL8, a multifunctional chemokine secreted by stromal cells in the inflammatory microenvironment, and its receptor CXCR1 have been extensively explored in wound closure[53] and tumorigenicity by us and other groups[54–58]. However, the effect of CXCL8-induced signaling in UC remains unclear. The highly significant roles of CXCL8 and GPCR signaling in the iHUCO transcriptome, with higher specificity of CXCR1 (binding only CXCL8 with high affinity) than CXCR2 (binding all ELR-CXC chemokines with high affinity[59]) led us to focus on the CXCL8/CXCR1 inflammatory mediator as a transducer of the G-protein-activating regulatory system[60]. Dual IF staining for CXCL8 ligand and CXCR1 receptor of the organoids revealed significantly greater ($p < 0.0001$) expression of both proteins in the epithelium and stroma of iHUC as compared with iHN organoids (Fig. 4g–j).

To summarize, the bulk transcriptomic analyses demonstrated that the iHUCO is a feasible in vitro model for UC. Furthermore, GO analysis identified the role of GPCR, CXCL8, and Rho signaling, a CXCR1/CXCL8 signal transducer and one of the multiple downstream effects of the CXCL8/CXCR1 interaction[61] in iHUCOs. Our dual IF staining confirmed upregulation of CXCL8 ligand and CXCR1 receptor in the epithelium and stroma of iHUC vs. iHN organoids.

**Single-nucleus analysis of the parental fibroblasts reveals a unique inflammatory signature**. To examine the similarities between parental fibroblasts and each developmental stage, we also conducted bulk RNA-Seq on UC and normal parental fibroblasts (GSE117345).

An unsupervised hierarchical clustering based on the Canberra distance showed that parental fibroblasts shared the highest level of similarity with the organoids rather than the other stages of development (Fig. 5a). Hypothesizing that this similarity originated from the organoid mesenchymal compartment, we used single-nucleus transcriptional profiling (snRNA-Seq) to better understand the cell types and transcriptomic heterogeneity in normal and UC parental fibroblasts ($n = 3$ each) and

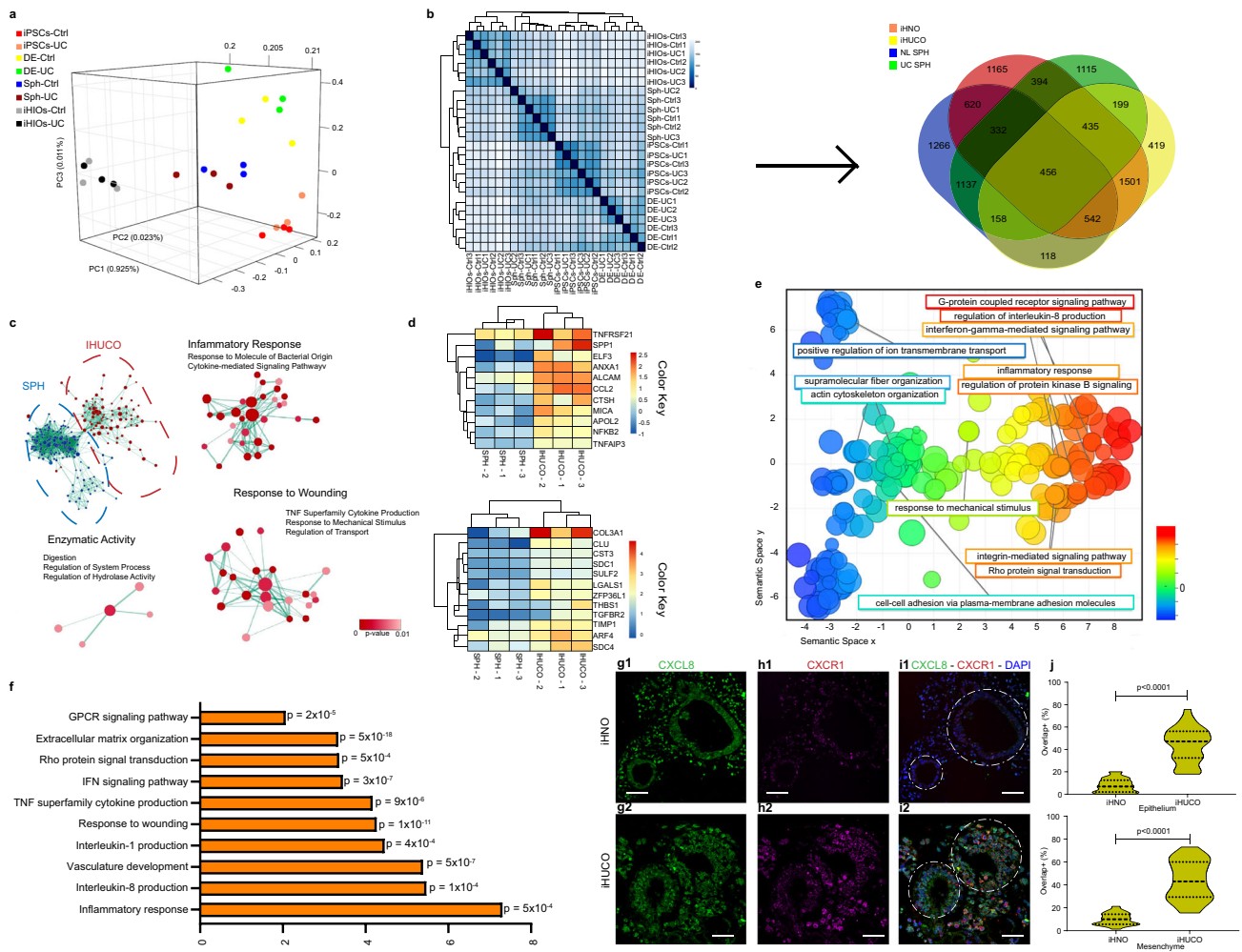

**Fig. 4 Bulk RNA-sequencing reveals UC signatures in iHUCOs. a** Principle component analysis (PCA) was conducted on normal and UC iPSC, DE, SPH, and organoid ($n = 3$ each). The first principal component (PC1) accounts for 92.5% of the variation in the data. **b** Spearman ranking was applied to cluster samples based on their similarity to generate a heatmap with the highest level of correlation in dark blue (left) and a Venn diagram (right) of differentially expressed genes (DEGs) for both UC and normal tissue developmental stages, including organoid, SPH, and DE. **c** DEGs in iHUCO vs. SPH were applied to conduct a functional network analysis (Cytoscape)[49] **d** Curated heatmaps of the key genes upregulated in the terms highlighted in panel **c**: inflammatory response (top), and wound healing (bottom). **e** DEGs in iHUCO vs. UC SPH were subjected to Gene Ontology (GO) analysis presented as REVIGO[50] scatterplots ($p < 0.001$). The terms extend along the X-axis based on similarity in the type of biological process (semantic space X); the difference in color highlights the variety of biological processes. Each circle represents a unique GO term. Circle size corresponds to the number of the genes associated with the GO term. **f** Highly enriched and significant GO terms in iHUCO shown as a bar graph for the enrichment scores. **g–i** Representative dual IF staining for CXCR1 (red) and CXCL8 (green) expressed in iHN and iHUC organoid epithelium and mesenchyme, along with **j** a violin plot for cells co-expressing CXCR1/CXCL8. All nuclei are stained for DAPI (blue). The unpaired, non-parametric two-sided Mann–Whitney U test was used to test for differences in CXCL8/CXCR1 overlap in expression between the iHNO and iHUCO epithelium vs. membrane. Scale bar = 25 µm. $n = 5$ iHNO and $n = 6$ iHUCO for IF. Source data are provided as a Source Data file.

compared them to their induced organoids (GSE152999). Single-nucleus profiling resulted in 12 and 9 predicted clusters in normal and UC parental fibroblasts, respectively (Fig. 5b). The list of top-expressed genes in each cluster is shown as marker plots for both normal and UC fibroblasts (Fig. 5c). Notably, IGFB5 was strongly expressed in all UC clusters shown in Fig. 5b. Overexpression of IGFBP5 has been linked to chronic inflammation via an increase in collagen synthesis, leading to mitogen-activated protein kinase (MAPK) and ultimately fibrosis[62–65].

Next, we used the known and recently reported list of markers in stroma of normal and UC colon tissues[66] to annotate the cell subtypes in both UC (Supplementary Fig. 4a) and normal (Supplementary Fig. 4b) fibroblasts and grouped them based on their identity. Results confirmed the presence of unique transcriptional subtypes such as inflammatory and fibrogenic[67]

in the UC but not in normal fibroblasts. The WNT2B+ and WNT5B+ subpopulations here are a component of the inflammation-associated fibroblasts noted by others in the UC population[66].

To identify the dominant biological processes enriched in UC, we applied the ranked list of DEGs in UC vs. normal fibroblasts to GOrilla, and summarized the results as a REVIGO plot (Fig. 5d). Gradual changes based on the GO term similarities along the X-axis identified highly significant terms ($P < 0.001$) involved in the inflammatory response and immune cell recruitment/migration, including "response to IL-6/IL-1 production" and "neutrophil degranulation," that lead to activation of biological processes such as chemokine-mediated processes and IFN-γ signaling (Fig. 5d, and Supplementary Fig. 4c). The genes playing crucial roles in enrichment of these terms were mainly

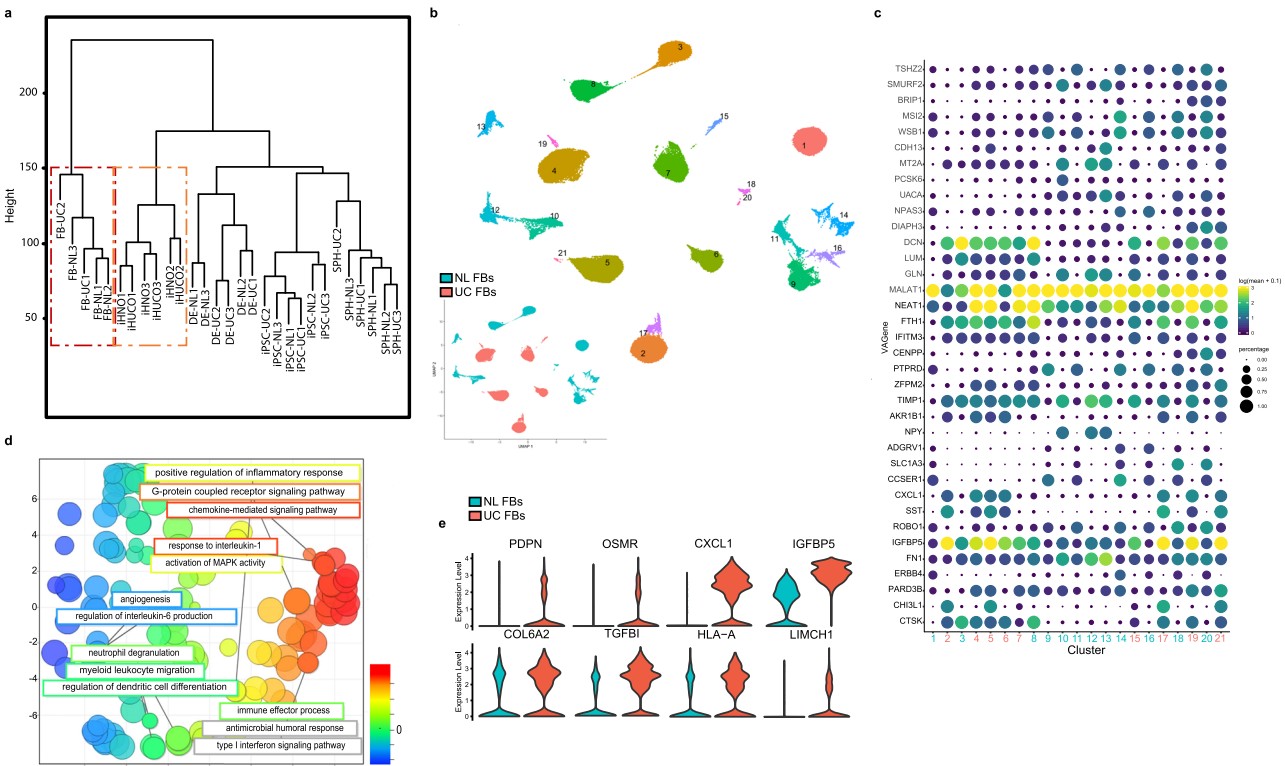

**Fig. 5 Single-nucleus analysis of the parental fibroblasts reveals a unique inflammatory signature. a** Dendrogram of the gene sets hierarchically clustered based on the Canberra distance. Parental fibroblasts ($n = 3$ each) and all 24 samples in different stages of development were included in the analysis. The red and orange boxes highlight the parental fibroblasts and induced organoids—they cluster next to each other and share the highest level of the similarity compared with the other stages of development. **b** UMAP for the predicted clusters in normal and UC fibroblasts. Inset, pink denotes UC fibroblast clusters, while turquoise denotes normal colon fibroblast clusters. **c** Marker plot of the top expressed genes in the predicted clusters in normal and UC fibroblasts (Fig. 5b). Dot color and size correlate with mean expression level and fraction of cells per cluster expressing each gene. Cluster number is colored to match parental cluster. Pink number indicates UC fibroblast origin while turquoise indicates normal colon fibroblast clusters. **d** Highly enriched GO terms in UC vs. normal fibroblasts shown as REVIGO[50] scatterplots ($P < 0.001$). The terms extend along the $X$-axis based on similarity in the type of biological process (semantic space X); the difference in color highlights the variety of biological processes. Each circle represents a unique GO term. Circle size corresponds to the number of the genes associated with the GO term. **e** Violin plots show normalized gene expression levels, contrasting normal and UC fibroblasts, for each gene. A smoothed and rotated kernel density curve is plotted. The width of the violin plot represents the abundance of the number of cells at each corresponding vertical axis gene expression value. A reverse "T" shape is an indication of a universal lack of expression among all cells. Here, raw gene expression levels for each cell in normal and UC fibroblasts were extracted, scaled, and normalized. These key genes are responsible for the immune/inflammatory response and ECM remodeling in UC fibroblasts, highlighting the upregulation of these genes in UC compared with normal fibroblasts $n = 30,894$ cells for normal fibroblast and $n = 67,654$ cells for UC fibropblast.

associated with inflammatory, chemotactic, and CD14+ subsets in UC, whereas the fibrogenic and POSTN+ subsets mostly identified the terms related to angiogenesis and ECM remodeling, in response to inflammatory cytokines. A selected number of top expressed genes in these GO terms are shown as violin plots in Fig. 5e to highlight their upregulation in UC when compared with normal fibroblasts. The roles of GPCR signaling, chemokine signaling, and regulation of the GPCR downstream pathways in UC fibroblasts were also highlighted in our GSEA analysis (Supplementary Fig. 4d). Specifically, genes such as podoplanin (PDPN), the lymphoid-tissue-like stromal marker, and oncostatin M receptor (OSMR) were expressed at an increased level[68]. Oncostatin M and its receptor are expressed in the stroma of patients with UC, and induces the expression of PDPN. Insulin-like growth factor binding protein-5 (IGFBP5)[64] and TGFβ1 have well-known roles in fibrosis[69]. MHC Class I proteins, here noted via upregulated expression of HLA-A[70] and LIMCH1[71], affecting actin stress fibers, have immune and wound-healing roles.

These results not only confirmed the colitic signature of UC fibroblasts at the transcriptome level and the essential role of CXCL8 and CXCL1 superfamily in stromal immunoregulation,

but also confirmed similar findings in a previous study showing the upregulation of the neutrophil chemoattractants CXCL1, CXCL5, CXCL8 in UC compared with non-inflamed tissues.

**High-resolution transcriptomic analysis of iHUCOs highlights their parental signature.** Several studies have highlighted the importance of interaction between colonic epithelium and the stromal compartment in both health and disease[72]. During UC, in addition to epithelial barrier dysfunction, mesenchymal and immune cell functions are disrupted. This disruption leads to the emergence of a population of activated mesenchymal cells that exacerbate disease severity by impairing epithelial cell maturation. Chronic inflammation also disrupts the cross-talk between the epithelium and immune cells, driving excessive local accumulation of immune cells and ultimately generating a pro-inflammatory mucosal state in colonic epithelium[28,73]. We used snRNA-Seq profiling to study the nuclear profile of iHUC and iHN organoids in the epithelial and stromal compartments (Figs. 6, 7).

UMAP analysis partitioned each of iHUCOs and iHNOs into 11 predicted clusters (Fig. 6a). A list of top markers in each

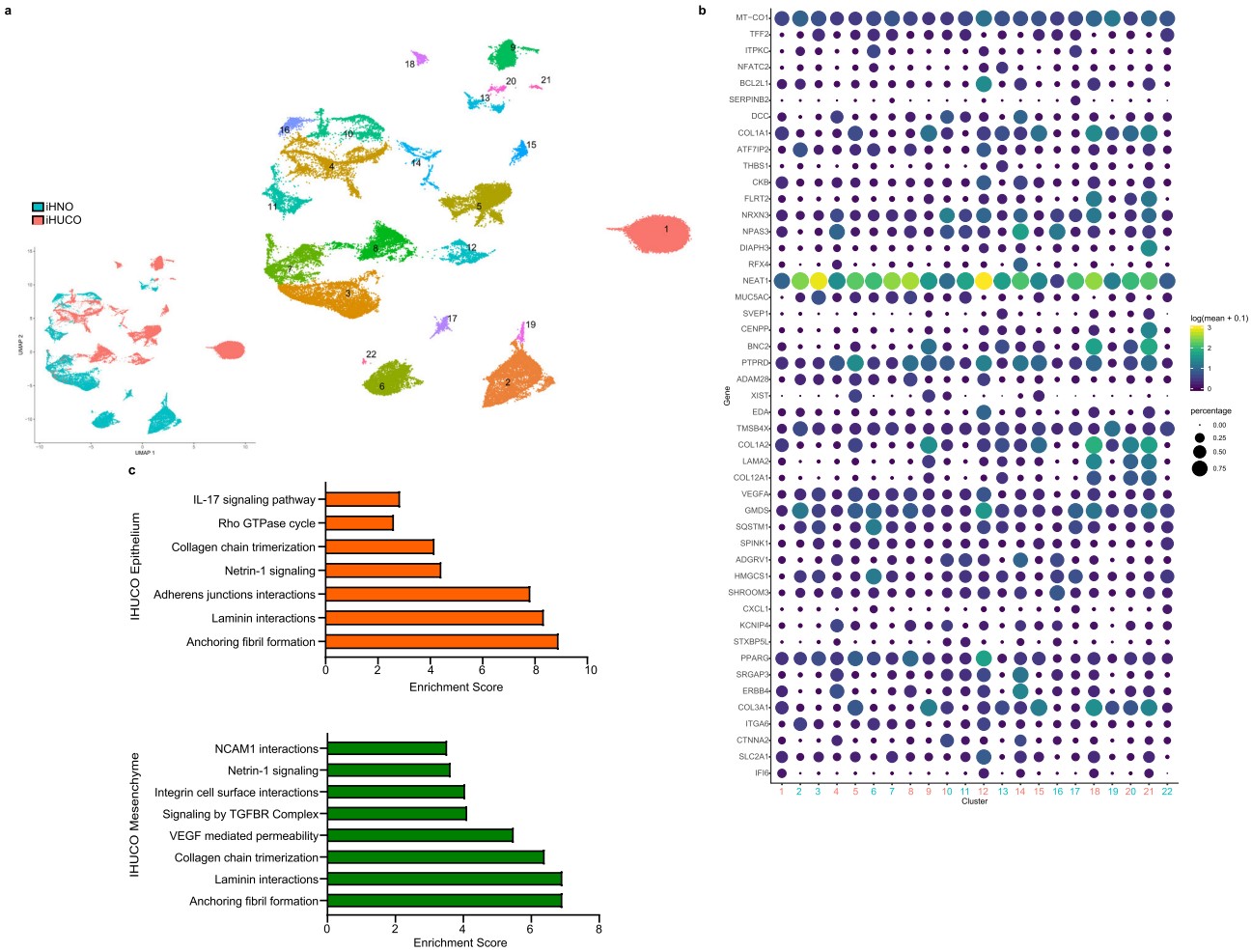

**Fig. 6 High-resolution transcriptomic analysis of iHUCOs highlights their parental signature. a** UMAP of the predicted clusters in iHNOs and iHUCOs (n = 3 each). Different colors are used to distinguish clusters. Inset, pink are iHUCO clusters, while turquoise are iHNO clusters. **b** Top expressed genes of the predicted clusters in iHNOs and iHUCOs shown as a marker plot. The color and size of dots correlate with mean expression level and fraction of cells per cluster expressing each gene, respectively. Cluster number is colored to match parental cluster. Pink number indicates UC fibroblast origin while turquoise indicates normal colon fibroblast clusters. **c** Reactome pathway analysis based on the DEGs in the epithelial and stromal compartments of iHUCOs vs. iHNOs, shown as bar graphs for the selected highly enriched and significant terms. n = 30,089 cells for iHNO and n = 43.420 cells for iHUCO.

cluster presented as a marker plot (Fig. 6b) highlights the overexpression of COL1A1, and COL3A1, in iHUCOs. The significance of these collagen genes lies in their relationship to the development of fibrosis, another hallmark of UC[74].

To understand the differences in signaling pathways between the epithelial and stromal components of the iHUCOs and the iHNOs, we performed a Reactome pathway analysis on the DEGs between the epithelial and stromal compartments of iHUCOs and iHNOs. This analysis confirmed a tight and interactive relationship between both cell compartments in iHUCOs (Fig. 6c), in particular, the epithelial and stromal coordination in nervous system development and neural migration as highlighted by Netrin-1 signaling. Further, Reactome terms such as "anchoring fibril formation" and "laminin interactions" with very high enrichment scores in both compartments highlight the vital role of interaction between the epithelial compartment and ECM in iHUCO development. Terms such as IL-17 signaling with downstream regulation of NFKB and MAP kinases, Rho-GTPase signaling, and adherens junctions were exclusively enriched in iHUCO epithelium whereas the hallmarks of fibrosis including TGFβ signaling, excessive collagen production, and VEGF mediated permeability were unique to stroma (Fig. 6c).

Further, using established markers, annotation of iHN and IHUC organoids showed that induced organoids consist of 5 main cell types: epithelial, stromal, immune, neural and other cells (Supplementary Fig. 5a). The presence of neural cells has been recently reported in human intestinal organoids[75]. Here we show a significant increase in these cell classes, as well as immune and stromal cells in the UC compared to the normal organoids (Supplementary Fig. 5a). Similar to parental fibroblasts, further annotation[66] of the cell subtypes in organoid epithelial and stromal compartments resulted in unique signatures in iHUCOs, correlating with the inflammatory and more immunogenic nature of UC (Supplementary Fig. 5b, Fig. 7a, b).

In iHUCO epithelium, we identified the presence of immature signatures (vs. mature in iHNO) in enterocytes and goblet cells along with the unique secretory signature of enteroendocrine cells[45,76] (Fig. 7b). These data are consistent with the reparative nature of UC and also with our in vivo observation of an increased number of chromogranin+ (enteroendocrine cell marker) cells in the crypts formed by iHUCOs but not iHNOs (Fig. 3h, i). Moreover, it is known that in the reparative state, adult stem cell markers such as Lgr5+ cells are repressed, and distinct intestinal stem cells act to replenish lost stem cells and contribute to crypt

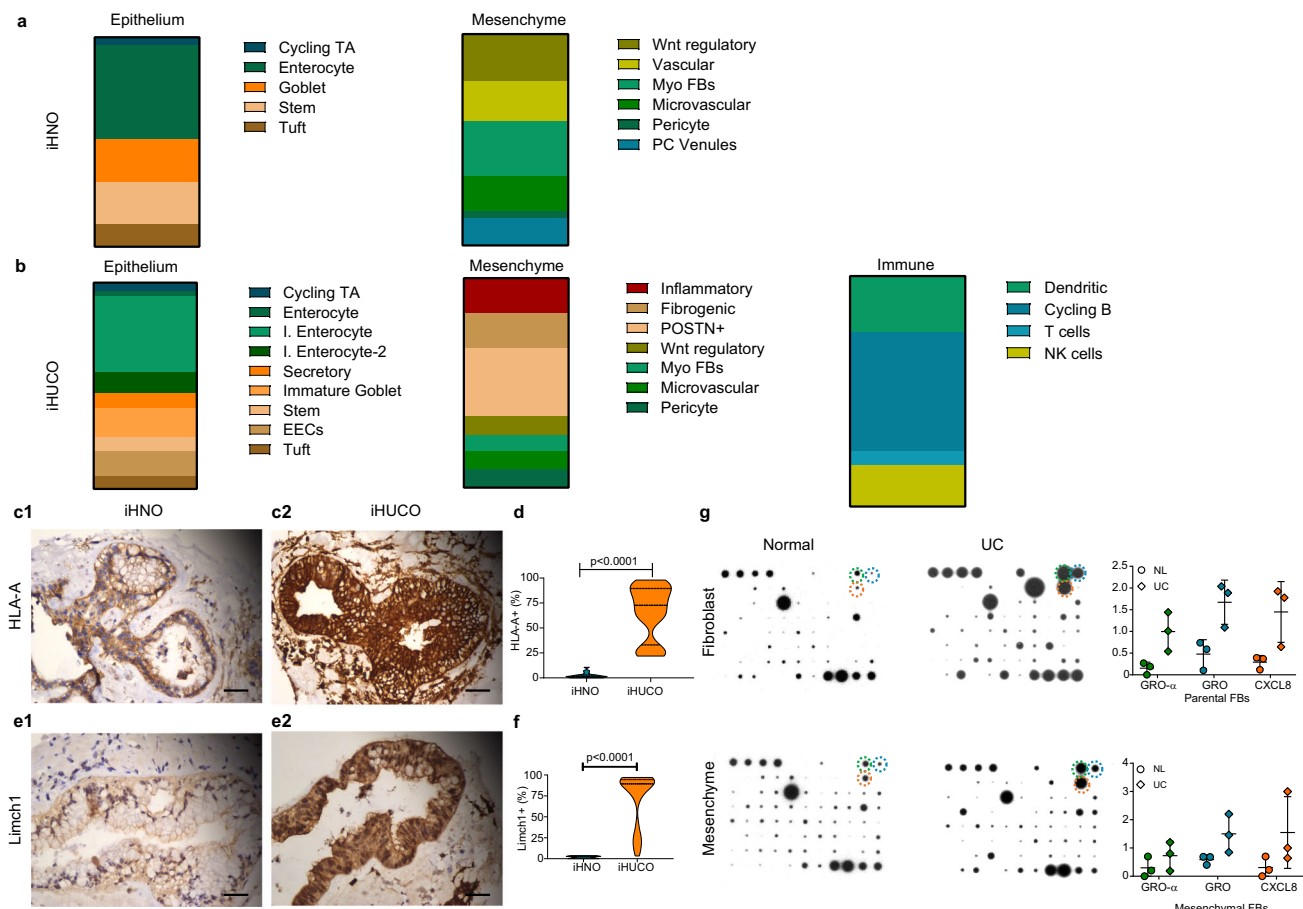

**Fig. 7 Single nucleus profile of iHNOs corresponds with their fibroblasts of origin. a** Proportion plots to demonstrate the identified sub-populations in iHNO epithelium and mesenchyme. Cycling TA = cycling transient amplifying cells; Myo FB = myofibroblasts; PC venules = pericyte venules. **b** Proportion plots illustrating sub-populations per cell type in iHUCOs. Cycling TA = cycling transient amplifying cells; I. enterocyte = immature enterocytes; I. goblet = immature goblet cells; Myo FB = myofibroblasts; cycling B = cycling B cells; NK = natural killer cells. **c–f** Representative IHC for HLA-A-expressing and Limch1-expressing epithelia in iHNO vs. iHUCO with violin plots showing a significant increase in expression of both proteins in iHUCOs compared with iHNOs. The unpaired, non-parametric two-sided Mann–Whitney U test was used to test for the difference in percentages between iHNO and iHUCO. Scale bar = 40 μm. $n = 5$ iHNOs (blue) and $n = 6$ iHUCOs (orange). **g** Representative cytokine arrays of normal and UC parental fibroblasts and organoid-derived mesenchyme with the quantification of three chemokines are increased in UC vs. normal: GRO-α (green), GRO $- α + β + γ$ (blue), and CXCL8 (orange). The median value is indicated in the center of the box plot. The whiskers above and below the plot represent 1 standard deviation (SD) above and below the mean of the data, respectively. Source data are provided as a Source Data file.

restoration[28,77]. A volcano plot for strongly up/downregulated genes in the iHUCO stem cell sub-population consistently showed suppression of Lgr5-expressing cells (Supplementary Fig. 5d) whereas genes such as SMOC2[78], which promotes matrix assembly and cell adhesiveness, were overexpressed. Our in situ hybridization for Lgr5 confirmed that this stem cell marker was suppressed in iHUCOs (Supplementary Fig. 5e, f).

Wound healing is a complex coordinated process, orchestrated by tight and dynamic interactions between different cell types and ECM components to produce successful repair. In the case of chronic wounds, this coordination is compromised and usually results in fibrosis (a hallmark of IBD) associated with excessive production of ECM components including collagen type-I[28]. Consistent with the significant signature of fibrogenic and POSTN+ subtypes in iHUCOs, IHC staining for both collagen type-I and periostin confirmed that these proteins are overexpressed in iHUCO but not iHNO ECM (Supplementary Fig. 5g, h).

Despite the limited number of immune cells (mostly cycling B) in iHNOs, we identified extended cycling B, T, and natural killer (NK) cell sub-populations in the immune compartment of iHUCOs. We also found dendritic cells (like the UC fibroblast

signature, Fig. 7b). Although the tight relationship between intestinal epithelial cells and dendritic cells is known, there is no report of direct interaction of dendritic cells with intestinal organoids in a co-culture system. However, a recent study examined the direct effect of dendritic cells on gastric organoids, confirming a steady state migration of these cells to the basolateral surface of gastric spheroids that is induced by the secretion of CXCL1 and CXCL8 chemotactic factors[79]. This study and our IHC data showing that that the immune-regulating MHC Class I, HLA-A[70], and Limch-1are significantly upregulated in iHUCOs but not iHNOs. (Fig. 7c–f) highlight the role of epithelial stromal cross-talk in adaptive and innate immune responses where dysregulation results in a pro-inflammatory state in the GI tract.

To further explore our observations, we isolated the mesenchymal compartment of iHN and iHUC organoids, verified by IF staining for vimentin (Supplementary Fig. 5i) and α-SMA (IF not shown) and the lack of CK19 (Supplementary Fig. 5i). We then performed a cytokine array and compared the secretome of isolated mesenchyme to parental fibroblasts (Fig. 7g). In those parental and resulting mesenchyme derived fibroblasts derived

from the most inflamed histology, the secretome revealed greater expression of CXCL8 (IL-8), as well as the IL-8-related chemotactic cytokines GRO-α (CXCL1), GRO-β-γ (now termed CXCL2, and CXCL3, respectively), activators of human leukocytes[80]) in both UC fibroblasts and mesenchyme compared with normal fibroblasts and tissue (Fig. 7g).

In conclusion, here we show that the colitic signature of UC fibroblasts results in a unique developmental signature in iHUC vs. iHN organoids. This highlights that the microenvironment, including stromal cells and ECM proteins, is crucial in iHUCO development. Both those UC parental fibroblasts and iHUCO-derived mesenchyme from the most inflamed tissues showed increased expression of CXCL8 and CXCL1 chemokine expression.

**Repertaxin attenuates in vitro the progression of the colitic phenotype in iHUCOs.** The upregulation of CXCL8/CXCR1 pro-inflammatory interaction in iHUCOs and their parental fibroblasts (Figs. 4, 5) led us to study the effect of repertaxin, a small molecule inhibitor of the CXCL8 receptor, CXCR1, on organoid development. In brief, we treated both UC and normal spheroids with repertaxin for 21 days during their development into organoids and compared the phenotypic characteristics to vehicle-treated (medium lacking repertaxin) control organoids. The results confirmed significantly lower expression of CXCR1 and CXCL8 and lower co-expression in the epithelium and mesenchyme of treated iHUCOs compared to controls (Fig. 8a–d).

Next, we examined the functional effect of repertaxin on iHN and iHUC organoid growth and morphology. Repertaxin treatment of iHNOs resulted in significantly ($p < 0.0001$) smaller organoids but did not significantly change their epithelial structure (Fig. 8e1, f1, g). In contrast, repertaxin treatment significantly affected the size and epithelial structure of iHUCOs, which changed from mainly stratified to columnar (Fig. 8e2, f2, h). In parallel, Ki67 expression was significantly lower in the epithelium of repertaxin-treated iHN ($p = 0.004$) and iHUC ($p < 0.0001$) organoids (Supplementary Fig. 6a–d). Repertaxin led to lower cytoplasmic and nuclear expression of β-catenin and E-cadherin in iHNOs than controls and a significantly higher number of cells with limited expression of β-catenin (Fig. 8i1, j1, k) and E-cadherin (Fig. 8m1, n1, o) on the plasma membrane ($p < 0.0001$). Conversely, the cytoplasmic and nuclear expression of both β-catenin (Fig. 8i2, j2, l) and E-cadherin (Fig. 8m2, n2, p) was greater in repertaxin-treated iHUCOs than in control organoids. We also studied the effect of repertaxin on RhoA expression. Although repertaxin did not significantly affect RhoA expression in iHNO (Fig. 8q1, r1, s), it significantly decreased expression in iHUCO epithelium (Fig. 8q2, r2, t). Claudin-1 expression did not significantly change in treated vs. control iHNOs (Supplementary Fig. 6e1, f1, g). However, claudin-1 expression significantly increased in iHUCOs after treatment with repertaxin (Supplementary Fig. 6e2, f2, g).

To functionally test the effect of repertaxin treatment on the epithelial barrier permeability, we used the microinjection technique[20], which allows real-time analysis of changes in permeability. Repertaxin did not significantly affect the epithelial permeability of iHNOs but did decrease iHUCO epithelial permeability (Supplementary Fig. 6h).

Therefore, CXCL8 receptor inhibition by repertaxin significantly attenuated development of the iHUCO colitic phenotype in in vitro. Repertaxin significantly affected the size and epithelial structure of iHUCOs, and it also modified the expression pattern of proteins regulating the adherens junction complex such that the pattern more closely resembled that of iHNOs. We

functionally validated these observations using the microinjection technique in real-time to show that although repertaxin treatment did not significantly affect epithelial barrier permeability in iHNOs, iHUCO demonstrated diminished epithelial permeability. A cartoon illustrating the effect of repertaxin on iHUCO epithelial intercellular junctions is shown in Fig. 8u.

**Repertaxin attenuates in vivo the progression of the colitic phenotype in iHUCOs.** To test the significance of repertaxin treatment in vivo, we studied the effect of repertaxin on the developmental progression of spheroids to organoids implanted subcutaneously in the dorsal flank of NSG mice (Fig. 9a). In brief, we encapsulated the spheroids in TS-HA hydrogel and implanted the bead subcutaneously. Mice were then treated daily for 21 days with either 20 mg/kg repertaxin or PBS (control). The overall volume after 21 days (measured twice weekly with calipers) was significantly greater in the control than in the repertaxin-treated (20 mg/kg) groups (Supplementary Fig. 6i). We confirmed this observation by H&E staining of the collected beads after 21 days (Fig. 9f–i). Consistent with our in vitro observation, repertaxin treatment affected iHUCO size and epithelial structure ($p < 0.0001$) whereas it had no significant effect on iHNO structure (Fig. 9h, i). Repertaxin also reduced the epithelial proliferation of iHUCOs as determined by Ki67 staining. However, it had no significant effect on iHNO proliferation (Supplementary Fig. 6j–m).

We also analyzed the beads for chemokine expression. Consistent with our in vitro data, CXCR1 and CXCL8 expression were significantly lower in both UC and normal organoids in mice treated with repertaxin than control organoids (Fig. 9b–e).

In addition, repertaxin treatment increased the membrane expression of both β-catenin and E-cadherin protein in iHNOs (Fig. 9k1, o1) whereas in iHUCOs, it resulted in greater cytoplasmic and nuclear expression of both (β-catenin, Fig. 9k2, m; and E-cadherin, Fig. 9o2, q). Further, repertaxin treatment had no significant effect on RhoA expression in iHNOs but resulted in significantly lower membrane and cytoplasmic expression of in iHUCOs (Fig. 9r–u). Claudin-1 expression in normal organoids also was not affected by repertaxin treatment, but it was significantly greater in repertaxin-treated iHUCOs, indicating greater regulation of tight junctions in iHUCO epithelium (Supplementary Fig. 6n–q). These data demonstrate that repertaxin treatment attenuated the colitic phenotype of iHUCOs not only in vitro but also in vivo in terms of epithelial structure, size, and changes of the epithelial intercellular junction.

## Discussion

The nature of UC is complex as it relies on interaction of different cellular compartments, and this has made it challenging to study the pathogenesis of colitis. Current therapeutic approaches in IBD mainly focus on the suppression of immune responses[81]; however, these therapies often fail, suggesting that the role of the epithelial and mesenchymal components in the development and progression of UC is not appreciated.

All models have limitations. Common in vivo rodent models or epithelial organoids as an in vitro model have advantages but do not adequately recapitulate the complexity and etiology of clinical UC[82]. In particular, to date, epithelial organoids have focused solely on the epithelium and do not include the role of micro-environment in colitis[83–85].

As an alternative, our induced in vitro model of human UC (iHUCO) includes both functional epithelial and stromal compartments and retains the complexity of the colitic phenotype of the tissue of origin, such as a compromised epithelial barrier, indicating a significant advance in UC modeling. Currently, in

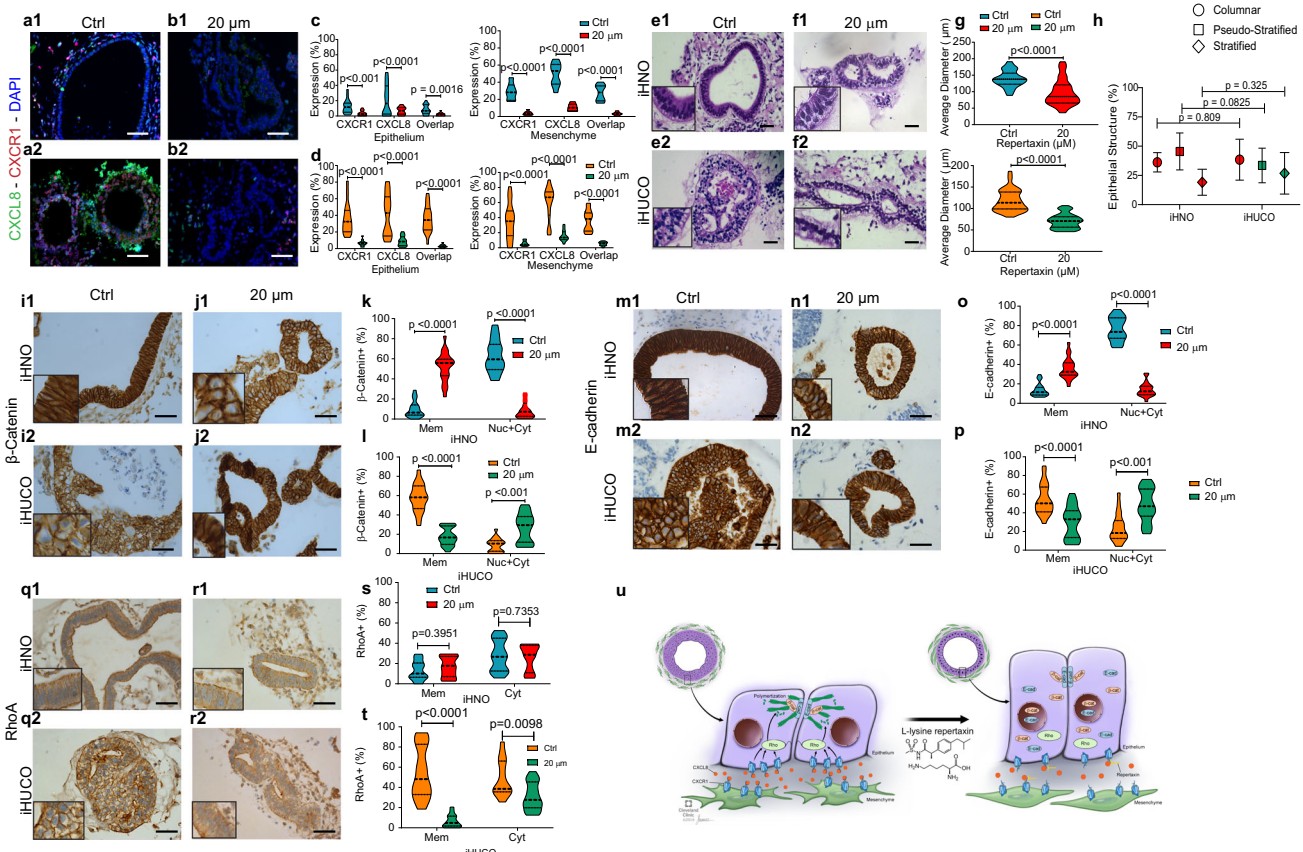

**Fig. 8 Repertaxin attenuates in vitro progression of the colitic phenotype in iHUCOs. a**, **b** Representative dual IF staining for CXCR1 (red) and CXCL8 (green) expressed in epithelium and mesenchyme of iHN and iHUC organoids in the absence or presence of repertaxin. **c**, **d** Violin plots of cells expressing CXCR1, CXCL8, and both (overlap), in the epithelium and mesenchyme of normal and UC organoids with or without (Ctrl) treatment with 20 μM repertaxin. **e**, **f** Representative H&E of iHNO and iHUCO with and without repertaxin treatment. **g**, **h** Summarized average diameter and percentage of epithelial structure type in organoids after 21 days, with and without repertaxin treatment. Violin plots in **g** compare iHNO control (ctrl) vs. 20 μm repertaxin treatment. In panel **h**, the epithelial structure is compared between iHNO and iHUCO in the presence or absence of 20 μm repertaxin. The median value is indicated in the center of the box plot. The whiskers above and below the plot represent 1 standard deviation (SD) above and below the mean of the data, respectively. **i**, **j**, **m**, **n** Representative IHC for β-catenin and E-cadherin in control and repertaxin-treated iHNOs and iHUCOs revealing altered cellular localization of the proteins after repertaxin treatment. **k**, **l**, **o**, **p** percentage distribution of the cells positive for β-catenin and E-cadherin according to the sub-cellular compartment; plasma membrane only (Mem) or membrane extended to cytoplasm and nucleus (Cyt + Nuc) with or without repertaxin treatment of the organoids. **q**, **r** Representative IHC for RhoA in the organoids shows the changes in cellular localization based on repertaxin treatment. **s**, **t** Violin plots for the cells expressing RhoA in control and repertaxin-treated organoids according to subcellular compartment: plasma membrane only (Mem) or cytoplasm (Cyt). **u** Schematic representation of the mechanism underlying the changes in epithelial intercellular junction in iHUCOs with and without repertaxin treatment. IHC Scale bar = 40 μm. IF scale bar = 25 μm. n = 5 iHNOs and n = 6 iHUCOs, respectively. For all comparisons, the unpaired, non-parametric, two-sided Mann–Whitney U test was used to test the difference between percentages of enumerated observations. Source data are provided as a Source Data file.

attempts to understand the mechanism by which inflammatory stroma can damage the integrity of the epithelial barrier, many studies try to develop a direct co-culture of intestinal epithelial cells, mesenchyme, and T cells. However, several layers of challenges, for example, histocompatibility issues, result in reduced mimicry of the in vivo dynamic between various cell types. Here, we introduce a human in vitro model that will allow us to identify and understand potential mechanisms behind the direct interaction of intestinal epithelial cells and stromal cells in a patient-specific manner. In particular, we identify morphologic transition, dysfunctional intercellular junctions and progression to early stages of fibrosis. High resolution transcriptional analysis of the IHUCOs and iHNOs demonstrated another level of discrimination. Increased levels of different transcripts confirmed findings of others including the presence of OSMR, PDPN, POSTN+, IGFBP5[64,65,68] which correlate with refractory disease and fibrosis. Further, key signatures demonstrating discrete cellular

infiltrates were identified that distinguished the iHUCOs from the iHNOs. These signatures included epithelial enteroendocrine cells, noted to have increased prevalence in UC[66] and not surprising with respect to our histological findings demonstrating increased numbers of cells as marked by chromogranin A. Another subpopulation within the stroma included fibrogenic and WNT2/5+ fibroblasts, which suggest the presence of inflammation-associated fibroblasts[66]. Further, we identified an immune population that was entirely unique to the iHUCOs. Consistent with others[86], B cells formed a dominant proportion of the immune subpopulation identified within the iHUCO cohort. Indeed, B cells have been the target of a small clinical trial in the past, which may not have been successful because the stromal subpopulation was not eliminated[87]. Nevertheless, these findings support the potential of our model to provide a comprehensive human model to manipulate immune and inflammatory regulators, naturally present in iHUCOs, to identify

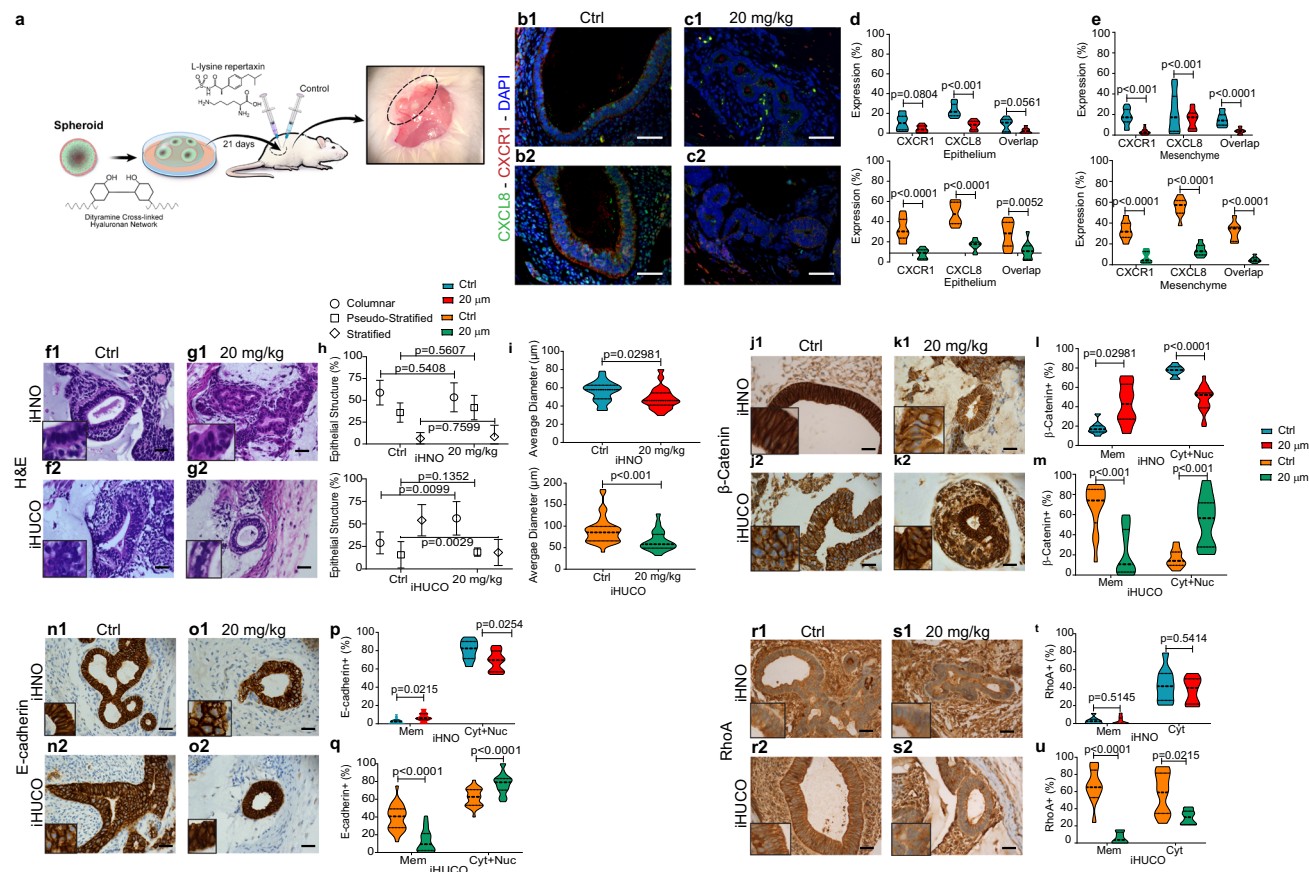

**Fig. 9 Repertaxin attenuates in vivo progression of the colitic phenotype in iHUCOs. a** Schematic representation of the in vivo repertaxin treatment study; spheroids encapsulated in TS-HA hydrogel were implanted subcutaneously in the dorsal flanks of immunocompromised NSG mice, which then received daily injections of repertaxin or PBS (control) for 21 days. **b, c** Representative dual IF staining for CXCR1 (red) and CXCL8 (green) in control (Ctrl; no treatment) and repertaxin-treated (20 mg/kg) implanted organoids. **d, e** Percentage distribution of cells positive for CXCR1, CXCL8 and both (overlap) in the epithelium and mesenchyme of normal and UC organoids in the presence or absence of repertaxin. **f–g** Representative H&E of control and repertaxin-treated organoids collected at 21 days. **h, i** Average organoid diameter and percentage of each epithelial structure type in the presence or absence of repertaxin. The unpaired, non-parametric, two-sided Mann–Whitney U test was used to test whether the percentage of cells expressing the specific morphology differed. The median value is indicated in the center of the box plot. The whiskers above and below the plot represent 1 standard deviation (SD) above and below the mean of the data, respectively. **j, k, n, o** Representative IHC for β-catenin and E-cadherin in control and repertaxin-treated organoids revealing the altered cellular localization of both proteins in iHN and iHUC organoids by repertaxin treatment. **l, m, p, q** Distribution of the epithelial cells positive for β-catenin and E-cadherin with or without repertaxin treatment. Results reported for the plasma membrane only (Mem) or membrane extended to cytoplasm and nucleus (Cyt + Nuc). **r, s** Representative IHC for RhoA in normal and UC organoids, confirming the effect of repertaxin on RhoA sub-cellular localization. **t, u** Percentages of cells expressing RhoA in control and repertaxin-treated organoids according to cellular compartment: plasma membrane only (Mem) or cytoplasm (Cyt). The unpaired, non-parametric, two-sided Mann–Whitney U test was used to test for the differences between all enumerated observations. IHC scale bar = 40 μm. IF scale bar = 25 μm. n = 3 each. Source data are provided as a Source Data file.

therapies to recover functioning epithelial cells in patients suffering from UC. With such a model, strategies for personalized medicine can be explored, and the mechanisms underlying the pathophysiology of human UC may be investigated in vitro.

Although the autocrine and paracrine functions of the CXCL8 chemokine and its receptor CXCR1 in the development of several types of cancer, including colorectal cancer, have been extensively studied[61,88] the downstream effect of this inflammatory interaction in UC development and progression remains unclear. Using iHUCOs, we provide evidence that shows overexpression of CXCL8/CXCR1 in UC results in a dysregulated adherens junction pattern in epithelial cells. Notably, CXCL8 lacks a murine homolog, which highlights the gap in the murine-based models and the functional importance of our model in identifying the role of CXCL8 receptor-mediated signaling in UC development and progression[89].

We also demonstrate the functionality of the model via response to chemical perturbation by repertaxin, a CXCR1

receptor small molecule non-competitive inhibitor. Exposure of both in vitro and in vivo organoid cultures to repertaxin reduced the expression of CXCL8 ligand and CXCR1 receptor and attenuated several aspects of the colitic phenotype, including a leaky epithelial barrier, suggesting that the pro-inflammatory interaction of CXCR1-CXCL8 compromises the epithelial barrier.

Our inducible organoid system provides the advantage of a patient-derived model to study the epithelial-mesenchymal and intestinal microenvironment interactions of human UC compared with the current murine models. Importantly, our model preserved the individual patient genetic background and thus, in the future, may be used to tailor IBD treatment[90] for regenerating colon and healing the damaged mucosa.

Finally, we demonstrated overexpression of CXCL8 and its receptor in UC patient tissues, validating the significance of our functional studies. Thus, using repertaxin to block this interaction may be a promising therapeutic strategy to diminish the chronic inflammatory symptoms of ulcerative colitis.

## Methods

**Patient population and tissue collection**. Tissue specimens were obtained from the distal colon of patients with UC who underwent biopsy or surgery for resection of UC at the Cleveland Clinic. None had strictures or sclerosing cholangitis, which are known risk factors for colitis-associated cancer. Patient demographics and clinical presentations are summarized in Supplementary Table 1. Patients with UC had disease for at least 2 years, therefore classified as chronic colitis. As UC patients often have pan-colitis, manifested as inflammation along the entire tract of the large bowel, normal tissue specimens were obtained from areas at least 10 cm away from gross tumors of patients presenting for resection of sporadic colon cancer. Pathology of retrieved tissues was confirmed by the Cleveland Clinic Pathology Department and approved under IRB 13-1159.

**Fibroblast isolation and characterization**. To isolate patient-derived fibroblasts[23], a segment of colonic tissue no larger than 0.5 cm³ was retrieved from the operating room, sterilized with antifungal-antimicrobial solutions for 15 min × 3, then minced in 500–1000 μL DMEM containing collagenase Type IA (1 mg/mL) in a round bottom 2 mL microfuge tube until the tissue could pass through a 1000 μL pipette tip. The minced tissue was transferred to a 50 mL conical tube with an additional 9 mL collagenase-containing media, then placed in a water bath at 37 ºC for 30 min. Collagenase activity was terminated by adding serum-containing medium to the minced tissue. The suspension was pelleted, re-suspended in 5 mL serum-containing medium and filtered through a 40-μm strainer. The filtrate was cultured on tissue culture plates in DMEM with 10% FBS and 1% pen/strep. The human colon fibroblast cell line CRL1541 was obtained from the American Type Culture Collection (ATCC). The identity of the isolated fibroblasts was confirmed by visual inspection for spindle-shaped cells, positive immunofluorescence (IF) staining for alpha-smooth muscle actin (α-SMA), and the absence of cytokeratin 19 (CK19) staining.

**Short tandem repeat analysis and mycoplasma testing**. A mycoplasma assay and short tandem repeat (STR) analysis (SupplementaryInformation Table 2) were conducted to verify the absence of mycoplasma and the unique origin of each fibroblast isolate, respectively. Once established, primary isolates of fibroblasts were also subjected to STR analysis to establish the unique genetic identity (Duke University DNA Analysis Facility and Genetica, Burlington, NC). Prior to experimental use, another STR analysis was performed to confirm the identity. Organoids derived from parental fibroblasts were also subjected to STR analysis, and the consistency in origin was validated (Supplementary Table 2).

**Reprogramming and iPSC characterization**. The CytoTune-iPS reprogramming kit (Supplementary Data File 1) was used according to the manufacturer's instructions to reprogram normal and UC fibroblasts (Supplementary Table 1) to induced pluripotent stem cells (iPSCs). The CRL1541 cell line (ATCC® CRL1541, CCD112CoN) was included to ensure that the reprogramming protocol and colonic organoid development were reproducible by other research groups using a commercially available cell line. In addition to the visual changes in the morphology of iPSCs and the parental fibroblasts, reprogramming was confirmed by using 1) Immunofluorescent staining for the known markers Tra-1-60 and Oct-4 to delineate pluripotency in human pluripotent stem cells, and 2) the human pluripotent stem cell identification kit (R&D Systems, Supplementary Data File 1) to validate the pluripotent capacity of iPSCs by trilaminar potentiality.

**Generation and patterning of induced human colonic organoids**. In brief, iPSCs were differentiated to definitive endoderm (DE) by adding DE induction medium (RPMI 1640, 2 mM L-glutamine, penicillin-streptomycin and 100 ng/mL[10,11] Activin A [Cell Guidance Systems]) supplemented with 0%, 0.2%, and 2.0% defined FBS for 3 consecutive days. On day 4, DE was treated with hindgut induction medium (RPMI 1640, 2 mM L-glutamine, 2% dFBS, penicillin-streptomycin) supplemented with 500 ng/mL FGF4 (R&D Systems), and 2 μM CHIR99021 (StemGent, Supplementary Data File 1). The medium was replaced daily for 4 days. To generate iHNOs and iHUCOs, spheroids were collected from 24-well plates, pooled, and plated in Matrigel (BD) with intestinal growth medium (Advanced DMEM/F-12, N2, B27, 15 mM HEPES, 2 mM L-glutamine, penicillin-streptomycin) supplemented with 100 ng/mL EGF (R&D Systems, Supplementary Data File 1), 100 ng/mL Noggin (R&D Systems, Supplementary Data File 1), and 500 ng/mL R-spondin 1 (R&D Systems, Supplementary Data File 1). The medium was changed twice weekly thereafter. Organoids were re-plated in fresh Matrigel every 7 days up to 21 days. In the case of mature organoids, this timeline extended to 42 days.

**Experimental design**. To confirm the reproducibility of the organoid generation and their phenotype, the previously described protocol of McCracken, K. W. et al.[11] was repeated by different investigators in more than 30 events, and the colitic phenotype in iHUCOs was applicable to the majority of organoids (~80%) for UC. Briefly, following reprogramming from colitic fibroblasts to pluripotency, confluent iPSCs were then subjected to directed differentiation to definitive endoderm using Activin A, and increasing percentages of defined fetal bovine serum over 3 days (none, 0.2%, 2%, in DE induction media). On day 4, hindgut

induction media was added, which includes Chiron and FGF4. On day 7, buds were pooled, harvested, and plated into 25 μL Matrigel beads. Heterogeneity between biological replicates matched observed phenotypes in the primary tissues. That is, when the chronic colitic tissue histology was less pronounced, so too was the organoid morphology. To avoid variations in phenotype, all generated organoids for the study were initiated from passage 0 originating from the iPSCs. To validate the consistency in results, all experiments were repeated on at least 3 separate occasions (independent experiments) and conducted on at least 5 independent biological replicates. A minimum number of 100 organoids per line were generated in each series of experiments.

**Immunostaining and microscopy**. Organoids were released from Matrigel beads through gentle washing in PBS followed by immediate fixation with 4% paraformaldehyde (PFA). The fixed organoids were concentrated into a 1.5% low melting point agarose plug prior to being embedded in paraffin. Standard conditions for antigen retrieval, including tris/borate/EDTA buffer (Discovery CC1, 950-500; Ventana) on a Discovery ULTRA automated stainer (Ventana Medical System Inc.), were implemented. Chamber slides with acetone-fixed organoids were used to perform immunohistochemistry (IHC)/immunofluorescence (IF) with the antibodies listed below. Images were captured on a Leica Microsystems microscope (version 4.3.0, Supplementary Data File 1). Cells expressing these proteins were counted manually (>500 nuclei/section/stain). IF on intact organoids was completed in suspension. Briefly, organoids were fixed for 1–3 h in 4% PFA, treated with blocking serum (5% serum in 1× PBS plus 0.5% Triton-X100) for 1 h prior to incubation with the primary antibodies overnight. The next day, the organoids were washed in PBS-0.5% Triton-X100 ×3 and exposed to the secondary antibody for 2 h. To counterstain for nuclei, organoids were suspended in a drop of VEC-TASHIELD mounting medium with DAPI and loaded on the coverwell imaging chamber for confocal microscopy. Images were captured on a Leica Microsystems confocal microscope (DM16000) using software version 2.7.3.9723, magnification ×40, and oil immersion. IHC on paraffin-embedded (4% PFA) fixed or snap-frozen sections of the primary tissues was performed with the antibodies listed in the Key Resources Table.

**Antibodies**. The following primary antibodies were used for staining in the study: anti-α-SMA (SIGMA anti-α-smooth muscle actin (SIGMA A5228 Clone 1A4, Lot Lot 074M4814V, dilution 1:200), anti-CK19 (Abcam ab8187, Clone EP1508Y, dilution 1:100), anti-Tra-1-60 (Abcam ab16288, dilution 1:500), anti-Oct4 (R&D System AF1759, Lot JTW0208081, dilution 1:500), anti-SOX17 (R&D System Part 967330, Lot KGA081041, dilution 1:10), anti-brachyury (R&D systems Part 967332, Lot KQPO315042, dilution 1:10), anti-Otx2 (R&D systems Part 967331, Lot KNO0415041, dilution 1:10), anti-FOXA2 [Santa Cruz Biotechnologies sc-6554 (discontinued) 1:100], anti-CDX2 (Biogenex MU392A-UC, clone CDX2-88, dilution 1:10), anti-vimentin (R&D systems MAB2105, Clone 280618, dilution 1:100), anti-Ki-67 (Novus Biologicals NB110-89717, Lot C1, dilution 1:1000), anti-SATB2 (Abcam ab51502, clone KQPO315042, dilution 1:200), anti-beta-catenin (BD Transduction Lab 610153, clone 14, dilution 1:200), anti-E-cadherin (R&D systems AF648, dilution 1:600), anti-RhoA (Abcam ab54835, clone 1B12, dilution 1:100), anti-CXCR1 (R&D systems MAB330, Clone 42705, dilution 1:800), anti-CXCL8 (Abcam ab7747, dilution 1:100), anti-CLDN1 (Abcam ab15098, Lot GR3248184-1, dilution 1:200), anti-HLA-A (Abcam ab52922, Clone EP1395Y, Lot GR258732-26, dilution 1:200), anti-Limch1 (Abcam ab96178, Lot GR15093-24, dilution 1:200)

The following secondary antibodies were used in the study: Alexa Fluor 488 donkey anti-goat (Thermo Fisher Scientific A11055, clone NA (polyclonal), dilution 1:600), Alexa Fluor 488 donkey anti-mouse (Thermo Fisher Scientific A21202, clone NA (Polyclonal), dilution 1:1000), Alexa Fluor 568 goat anti-rabbit (Thermo Fisher Scientific A11036 clone NA (Polyclonal), dilution 1:1000), Alexa Fluor 568 donkey anti-mouse (Thermo Fisher Scientific A10037 clone NA (Polyclonal), dilution 1:200)

**RNA isolation (Bulk RNA-Seq)**. Organoids were harvested after 21 days and washed with cold PBS. Total RNA was extracted using miRNeasy kits (Qiagen). RNA purity and integrity was assessed using a Bioanalyzer (Agilent). All samples had RNA integrity >9.5. Total RNA was processed for next-generation sequencing using TruSeq Total RNA kits (Illumina). Total RNA of each sample was depleted of ribosomal RNA using biotinylated oligomers, combined with Ribo-Zero rRNA removal beads and fragmentation of RNA using divalent cations at elevated temperatures. Cleaved RNA fragments were copied into double-stranded cDNA using (sequentially) reverse transcriptase, random primers, DNA polymerase, and RNase H. After ligation of the sequencing adapters, the cDNA products were purified.

**Bulk RNA-Seq processing**. RNA library construction and RNA sequencing was performed by the Case Western Reserve University School of Medicine Genomics Core, using an Illumina HiSeq2500 platform. Paired end 100-bp sequencing reads generated from this platform were assessed for quality control and trimming using FastQC, version 0.11.5[91] and multiqc, version 0.7[92]. The QC analysis identified no major issues. After removing the Illumina universal adapter sequence by Cutadapt[93], we used the software package STAR version 2.5.3a[94] and gene annotation from Gencode[95] to align QC-ed reads to the reference transcriptome HG19. The

total numbers of mapped reads to genes (read count) were counted using feature counts[96], and $\log_2$ (RPKM) values were calculated.

Unsupervised analyses including principal component analysis (PCA) and hierarchical clustering (HC) were performed to visualize the similarities and the distinctions among all the groups. In brief, the first three principle components were calculated using $\log_2$ (RPKM) values and 'prcomp' function in R and were plotted into a 3-dimensional space using the 'rgl' package. The $\log_2$ (RPKM) values were also subjected to HC analysis, using the complete agglomerative method and 'hclust' function in R. Next, the read count values and DEseq2 version 3.8 were used to identify differentially expressed genes (DEGs) and the gene signatures in developmental progression of both UC and normal groups[97]. To improve the reliability and accuracy of DEGs,-genes with minimal expression were filtered to ascertain the noise/error removal (genes less than 10 mapped reads in more than half of the total samples). Student's t-test and Benjamini-Hochberg False Discovery Rate (FDR) tests were performed for each of the DEGs in comparison groups.

The raw DEGs for iHUCO vs. spheroid were also subjected to Gene Set Enrichment Analysis (GSEA). Each gene of the DEGs was scored as 1/ (adjusted P-value * sgn ($\log_2$ (fold change)) where "sgn" stands for a sign function. The adjusted P-value and fold change values were extracted from the DEG analysis results. The scored genes were sorted in descending order and converted to .rnk file as an input for GSEA preranked[98]. KEGG[99], Reactome[100], BioCarta[101], and Biological Process in Gene Ontology[102] were used as gene set databases. The summary of the GSEA report for iHUCOs and spheroids was applied to create an enrichment map in Cytoscape[49]. To identify the biological processes enriched in iHUCOs vs. spheroids, we applied Analysis and Visualization (GOrilla), and Reduce and Visualize Gene Ontology (REVIGO). Specifically, the single ranked list of DEGs in iHUCO ($P \le 0.001$) was applied to GOrilla, identifying the enriched GO terms in biological process ontology. The output was shown as a 2-dimensional graph in REVIGO for the terms of interest (Supplementary Data File 1).

All heatmaps were generated using RPKM values in R, version 3.5.1 and the R packages gplots, RcolorBrewer, and pheatmap.

**Nuclear isolation (snRNA-Seq)**. To isolate nuclei, we followed the protocol suggested by Habib et al.[103] for both fibroblasts and induced organoids. Cultured fibroblasts at lower passages (<10) or 30 days old induced organoids were collected, washed, suspended in ice-cold EZ PREP buffer (see Key Resources Table), and homogenized using a glass dounce tissue grinder (×20 for each loose and tight pestle) and incubated on ice for 5 min, with an additional 2 mL of ice-cold EZ PREP. Next, nuclei were centrifuged at $500 \times g$ for 5 min at 4 °C, washed with ice-cold EZ PREP and filtered through a 40-μm cell strainer. After centrifugation, the nuclei were washed in 4 mL nuclei suspension buffer (NSB; consisting of 1× PBS, 0.01% BSA and 0.1% RNase inhibitor) and filtered through a 20-μm cell strainer, and then we proceeded with the 10x Genomics pipeline for sequencing.

**SnRNA-seq data processing and analysis**. Using Cell Ranger 3.1.0 from 10× Genomics, raw snRNA-seq reads were processed for alignment and generating feature-barcode gene expression count matrices. A GRCh38 pre-mRNA reference genome was built and adopted to map the sequence reads. Each biological sample went through the same procedure of alignment in parallel. After individual alignment, samples were aggregated using the Cell Ranger aggr function to create the integrated data and facilitate downstream comparison. We retained genes that had expression in at least 10 nuclei and nuclei that have expression of at least 200 genes, using Seurat 3.1.2 package[104] for analyses. To ensure high quality, we filtered out the nuclei expressing high level (>10%) of genes that mapped to the mito-chondrial genome. After filtering, data normalization and scaling were performed using Monocle 3 package[105,106] in R programming language, version 3.6.2.

After preprocessing and data quality control, a total of 65,963 and 67,856 nuclei were retained in normal and UC fibroblasts, respectively. Similarly, 30,819 and 44,185 nuclei were retained in iHNOs and iHUCOs. To compare UC with normal fibroblasts and organoids, we extracted the nuclei based on their type (diseased vs. normal) out of the integrated data and followed the normalization and data scaling steps before downstream analysis. In each group, we kept the first 50 principal components for data normalization using Principal Components Analysis (PCA) for Monocle 3 to for calculations. Next, the uniform Manifold Approximation and Projection (UMAP) technique was used for dimensionality reduction and visualization. Post-alignment data were read into Seurat for data normalization and scaling. Using the Wilcoxon Rank Sum test integrated in Seurat, we identified the DEGs comparing normal and UC fibroblasts. DEGs were defined as those genes that showed statistical significance and at least 0.25-fold change in log scale. Adjustment was made for multiple testing, and the FDR was controlled. Similarly, we identified DEGs comparing iHUCOs with iHNOs (see Supplemental information). Once we clustered the cells, we identified the marker genes for each cluster, using the integrated regression framework in Monocle 3 for normal and UC fibroblasts or induced organoids. Based on the list of marker genes reported by Smillie et al.[66] we annotated the cells using the SCINA algorithm[107] and each cell cluster was annotated by its dominating cell type. To identify the biological processes enriched in UC vs. normal fibroblasts, we applied the list of DEGs to GOrilla and REVIGO. Specifically, the single ranked list of DEGs in iHUCO (p-value <= 0.001) was applied to GOrilla, identifying the enriched GO terms in biological process ontology. The output was shown as a 2-dimensional graph in

REVIGO for the terms of interest. We then performed the Reactome pathway analysis on the iHUCO epithelial and stromal compartments. The list of DEGs in each compartment were applied to Gene Ontology Resource and processed for Reactome pathway analysis.

**Isolation of organoid-derived mesenchyme**. The 21-day normal and UC orga-noids were dissociated in collagenase type IA (2 mg/mL) at 37 °C for 20 min. Collagenase activity was terminated by adding serum-containing medium to the mixture. The pellet was re-suspended in serum-containing medium and filtered through a 40-μm strainer. The resulting suspension was plated in DMEM with 10% FBS to isolate organoid-derived mesenchyme. Visual inspection for the spindle shape, and positive immunofluorescence for the mesenchymal marker, vimentin, and negative stain for the epithelial marker CK19 were used to confirm the mesenchymal identity of the cells. Mesenchymal isolates were propagated and used for the further analysis.

**Human cytokine array C5**. To perform the cytokine arrays from the conditioned media of parental fibroblasts and organoid-derived mesenchyme ($N = 3$ each), $10^5$ cells were cultured in 1 mL of DMEM base medium for 24 h in the absence of serum. The conditioned media were harvested and subjected to quantification using the human Cytokine Array C5 (RayBiotech) according to the manufacturer's instructions.

**Measurement of epithelial barrier permeability**

*Microinjection*. Microinjections were performed using a modified protocol from Leslie et al.[108] and described in[20]. Briefly, organoids were injected using thin-wall glass capillaries and a P-30 micropipette puller (Sutter Instruments, Novato CA). Pulled microcapillaries were mounted on a Xenoworks micropipette holder with analog tubing attached to a 10-mL glass syringe filled with sterile mineral oil. Fine control of the micropipette was achieved using a micromanipulator (Narishge International Inc.) and microinjection was completed under ×1–2 magnification on an SX61 stereo dissecting scope (Olympus).

*FITC-dextran permeability assay*. For epithelial permeability assays, organoids were microinjected with 4 kDa FITC-dextran suspended in PBS at a concentration of 2 mg/mL as described previously[20] using the microinjection system detailed above. Images were collected at 30 min intervals at ×4 magnification on an Olympus IX71 epifluorescent microscope using a CellSens live cell imaging system (Olympus Corp). Cultures were maintained in 5% $CO_2$ throughout the imaging time course. The relative fluorescent signal was quantified from the collected images using ImageJ and custom macros following the protocol previously described[20].

**In vitro repertaxin experiments**. Matrigel encapsulated iHUCOs vs. iHNOs were treated with 20 μM repertaxin vs. control (PBS) for 21 days. Beads were harvested, Matrigel dissolved, and placed into agarose plugs. The agarose plugs were then embedded into paraffin blocks, stained with various antibodies of interest, and enumerated. A minimum of 500 cells/condition/type were enumerated. Assays were completed in at least triplicate using $N \ge 3$.

**Animals**. For the subcutaneous implantation and omental transplantation experiments, 4-week-old to 6-week-old and 12-week-old to 27-week-old immune-deficient NOD.Cg-PrkdcscidIl2rgtm1Wjl/SzJ (NSG) mice were used, respectively. All animal experiments were performed with the approval of the Cleveland Clinic IACUC (Protocol 16-1666). All mice were housed in the animal facility at the Cleveland Clinic Lerner Research Institute in accordance with NIH Guidelines for the Care and Use of Laboratory Animals. Animals were maintained on a 12 h light-dark cycle with access to water and standard chow *ad libitum*. The animals were identified by numbered cages and by ear punches. Ambient temperature was maintained at 71 °F, with 45–50% humidity.

**Omental transplantation of organoids**. The 21-day normal and UC organoids (3 independent biological replicates) were removed from Matrigel and encapsulated in tyramine-based hyaluronan (TS-HA) hydrogel (a kind gift from Dr. A. Calabro) for a total of 10 beads (20 organoids per bead) per group (technical replicates). The hydrogel was prepared at 4 mg/mL in PBS and was activated by horseradish per-oxidase at a ratio of 500:1. Equal numbers of spheroids (between control and treatment groups) were suspended in 25 μL activated hydrogel and cross-linked by exposure to 0.03% hydrogen peroxide (1 μL). Cross-linked hydrogel beads were transferred to 24-well plates and incubated in organoid medium for 2 h, then the medium was replaced with fresh organoid medium to assure removal of residual chemicals. We adapted a transplantation protocol[43] to xenograft each seeded bead surgically within the omentum of a host NSG mouse. The omentum was exposed from the mouse peritoneum, wrapped around the seeded scaffold, and tied with a 6-0 prolene purse string suture to securely contain the bead. The wrapped unit was then reduced back into the peritoneal cavity and the surgical incision was sutured closed. Post-operative treatment included buprenex, 0.1 mg.kg twice daily for 48 h. After 90 days, host mice were euthanized, beads were harvested and fixed in 4% PFA prior to embedding in paraffin blocks for histologic evaluation.

**Subcutaneous implantation of spheroids**. TS-HA hydrogel (a kind gift from Dr. A. Calabro) was used to encapsulate the intestinal spheroids preceding the subcutaneous implantation[109]. The hydrogel beads were prepared as above for the omental encapsulation experiments. The day after, the beads were subcutaneously implanted in the right dorsal flank of NSG mice[23]. The growth was measured twice per week using calipers. On day 2 post-implantation, repertaxin, 20 mg/kg vs. control diluent (PBS) was injected in rotating sites near the site of implantation for the duration of the experiments. Tumor volumes were calculated using the formula: volume = width$^2$ × length (mm, length as the longer measurement). The beads were harvested 21 days after implantation for further characterization.

For these experiments, blinding occurred at two levels: the investigator implanting the beads was blinded as to the contents of the beads. The investigator doing the treatments and the measurements was also blinded. The investigator performing the enumeration was likewise blinded.

**Quantification and statistical analysis**. The data presented as individual points or violin plots represent the percentage of positive cells for the target protein in $n = 5$ for normal and $n = 6$ for UC (biological replicates). A minimum of 500 nuclei per group in epithelia and 100 nuclei per group in mesenchyme (if applicable) were evaluated. Each independent biological sample was comprised 50–70 organoids pooled together per experiment for further analyses. Because the data were not normally distributed, an unpaired, non-parametric Mann–Whitney t test in GraphPad Prism 8.0 was performed for comparison of two groups. For the microinjection assay, unless otherwise indicated in the figure legends, differences between experimental groups or conditions were evaluated using a non-parametric Wilcoxon rank-sum test. All statistical analyses in the microinjection assay were conducted using R, version 3.5.3 (2019-03-11), and the plots were generated using the R package ggplot2.

**Statistical power, single nuclear analyses**. Based on the sequencing depth and the abundance of cells in this snRNA-seq, sufficient statistical power was achieved to detect differentially expressed genes and cell clusters. We had 5 biological samples for the normal condition and 6 biological samples for the colitis condition. These yielded a total of ~174,000 cells for all organoids combined. On average, we observed 23,871 number of reads per cell before normalization. We conducted simulation studies using the state-of-the art power assessment tool POWSC[110], which modeled expression level using a mixture of zero-inflated Poisson (ZIP) and log-normal Poisson (LNP) distributions. For small group comparisons with only 2k cells per sample, the power ranges from 0.788 to 0.985, with a marginal power of 0.922. The range of power comprehensively reflected the different strata of average reads. In all our single-cell analyses, we have achieved higher power, as the cell number is already much higher than in the simulation setting.

**Reporting summary**. Further information on research design is available in the Nature Research Reporting Summary linked to this article.

## Data availability

The next generation sequencing (NGS) datasets from this study have been deposited in the Gene Expression Omnibus repository (GEO), series accession number: GSE117345) and GSE152999. The rest of data that support the findings of this study or further information and requests for reagents may be directed to the corresponding author upon reasonable request (huange2@ccf.org). Source data are provided with this paper.

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

## Acknowledgements

We thank the members of Huang laboratory for their help with the project, Cassandra Talerico, PhD, for her help with scientific editing, Anthony Calabro, PhD, for the generous transfer of TS-HA hydrogel technique, and Haiyan Lu, MD, as a pathologist. This project was supported by the American Society of Colorectal Surgeons Limited Project Grant 100, NIH R01 CA142808, NIH R01 CA157663, NIH U01 CA214300, and NIH R01 CA237304. We thank the Cleveland Clinic Lerner Research Institute Imaging Core, the Pluripotent Stem Cell Facility at University of Florida (Nao Terada, PhD), This work also was supported by the Cancer Tissue Engineering Collaborative (U01CA214300), the Intestinal Stem Cell Consortium (U01DK103141), a collaborative research project funded by the National Institute of Diabetes and Digestive and Kidney Diseases and the National Institute of Allergy and Infectious Diseases (NIAID) to J.R.S. and by the NIAID Novel, Alternative Model Systems for Enteric Diseases (NAMSED) consortium (U19AI116482) to J.R.S.

## Author contributions

S.K.S. and E.H. conceived the study and experimental design. S.S., S.K.S., and R.F. established the techniques under supervision of the J.R.S. lab. S.K.S., E.H., X.S., J.R.S., R.K.D., and S.F. contributed to writing or editing the manuscript. S.K.S., S.F., and J.S. performed and analyzed the experiments. S.K.S. and E.H. performed omental transplantation and xenograft experiments. D.H. and J.R.S. performed the study of epithelial barrier permeability. B.H., Y.Y., H.F., T-H.H., and Y.N. performed bioinformatic analyses. M.C. performed the observations on the pathology of tissues/organoids.

## Competing interests

The authors declare no competing interests.
