## [Peer Review File · Nature Communications]

Reviewers' comments:

Reviewer #1 (Remarks to the Author):

In this work the authors reprogram colonic fibroblasts isolated from UC patients to iPSCs, followed by directed differentiation to induced human UC organoids (iHUCOs). They then compare these iHUCOs to induced human normal organoids (iHNOs). They find iHUCOs include both epithelium and mesenchyme and display some features of colitis associated epithelia such as aberrant proliferation, reduced goblet cells/mucus secretion, and compromised cellular tight junctions in the epithelial barrier. Via a transcriptional analysis they find the inflammatory chemokine CXCL8 was elevated in transcriptome and mesenchymal secretome of iHUCOs. Treatment of the cultures with the CXCL8 antagonist, repertaxin attenuated the colitic features in iHUCOs both in vitro and in a xenograft model in vivo.

Overall this is a nicely executed study which may prove of great use to the field. A flaw in the work is that there is no clear molecular explanation as to why colitis derived fibroblasts should drive the phenotypes observed in their models. Although CXCL8/CXCR1 over expression may explain part of this, further work is required to increase the confidence in the findings and general utility of their model.

Major:

- 1) At present it is not clear why 80% of their iHUCOs display colitis associated features and also what primary features of the starting material confers the colitis phenotype detected in the differentiated cultures. A more in-depth molecular analysis of the starting fibroblasts should be undertaken in health and UC. Recently single cell studies and others have highlighted the existence of "activated fibroblasts" in IBD expressing defined marker genes that hallmark this pathogenic subset including PDPN, periostin, IL33, CCL19, CCL21, OSMR, Lysyl oxidases, LIGHT, IL11R for example (Kinchen et al., Cell, 2018; Smillie et al., Cell, 2019; Martin et al., Cell 2019; West et al., Nat Med, 2017). It is possible this particular subset forms the starting material in the cultures that display the colitic phenotype as opposed to those that do not. It would strengthen the manuscript for the authors to define features of the originating fibroblasts and correlate these with colitis phenotype of subsequent iHUCOs. It may be only specific fibroblast states such as "activated fibroblasts, which are more expanded in more severely active IBD, are linked to development of the phenotype they observe. This will be important in increasing the reproducibility and general utility of their model system
- 2) Can they better characterise the mesenchymal cells present in the differentiated model in the light of recent single cell analysis ie do their mesenchymal cells display niche features such as expression of Gli1/F3/Sox6 atypical Wnts/BMPs etc that are dysregulated in iHUCOs. This would better define the nature of cross-talk between fibroblasts and epithelial cells in their cultures in health vs UC that give rise to the colitis phenotype.
- 3) Can the authors provide some more characterisation of the epithelial cells produced in this system. Can they detect other subtypes apart from goblets and EECs? Can they detect dysregulation of epithelial expressed defence pathways and an interferon response in iHUCOs?
- 4) In the discussion it would be useful to discuss the relative advantages of their model over non-iPS derived human intestinal epithelial organoids for studying barrier function in health and IBD in the light of their findings.

Minor:

Line 45 "leading to colectomy" would be better than "leading to ultimate resection of the colon and rectum"

Reviewer #2 (Remarks to the Author):

The manuscript „Induced organoids derived from patients with ulcerative colitis recapitulate colitic reactivity” by Sarvestani et al. developed an induced human ulcerative colitis-derived organoid model using induced pluripotent stem cells. Sarvestani et al. show that iHUCOs recapitulated histological and functional features of primary colitic tissues, and that the CXCL8/CXCR1 axis was overexpressed in UC, but not in NL organoids. Furthermore, Sarvestani et al. show that inhibition of CXCL8 receptor decreased the progression of UC phenotypes in vitro and in vivo.

The work of Sarvestani et al. gives new insights into the pathogenesis of UC by aid of a patient-derived in vitro model recapitulating UC, which makes this study of major interest in the field of ulcerative colitis as there is a major interest in understanding of UC pathogenesis. The manuscript describes a novel in vitro model of induced organoids from patients using induced pluripotent stem cells. The manuscript is well written and set up in a logical manner. The subject of the article is very suitable for publication in Nature Communications.

Major comments

- I suggest to increase number of patient tissue specimens and induced organoids for studies to at least n=10.
- I suggest that the quantification of IHC and IF is performed by two independent individuals.

Point-by-point Response to Reviewer #1

Reviewer 1 commented that “Overall this is a nicely executed study which may prove of great use to the field. A flaw in the work is that there is no clear molecular explanation as to why colitis derived fibroblasts should drive the phenotypes observed in their models. Although CXCL8/CXCR1 over expression may explain part of this, further work is required to increase the confidence in the findings and general utility of their model”.

Response: We greatly appreciate the reviewer’s enthusiasm and thoughtful observations. Rigorous efforts have been taken to address the deficits noted by the Reviewer, as detailed below.

Major comments

- 1) At present it is not clear why 80% of their iHUCOs display colitis associated features and also what primary features of the starting material confers the colitis phenotype detected in the differentiated cultures. A more in-depth molecular analysis of the starting fibroblasts should be undertaken in health and UC. Recently single cell studies and others have highlighted the existence of “activated fibroblasts” in IBD expressing defined marker genes that hallmark this pathogenic subset including PDPN, periostin, IL33, CCL19, CCL21, OSMR, Lysyl oxidases, LIGHT, IL11R for example (Kinchen et al., Cell, 2018; Smillie et al., Cell, 2019; Martin et al., Cell 2019; West et al., Nat Med, 2017). It is possible this particular subset forms the starting material in the cultures that display the colitic phenotype as opposed to those that do not. It would strengthen the manuscript for the authors to define features of the originating fibroblasts and correlate these with colitis phenotype of subsequent iHUCOs. It may be only specific fibroblast states such as “activated fibroblasts, which are more expanded in more severely active IBD, are linked to development of the phenotype they observe. This will be important in increasing the reproducibility and general utility of their model system.

Response: *We thank the reviewer for this critical comment. For more in-depth analyses of the parental UC vs. normal fibroblasts, we took advantage of single nucleus RNA-seq (snRNA-Seq) technology with the ability to capture the rare and difficult-to-distinguish cell subpopulations as well as to eliminate artifacts of dissociation-induced transcriptional stress. Our results led to brand new datasets (**Fig. 5 and supplementary Fig. 4**) that revealed the presence of the unique cell subpopulations in UC but not in normal*

fibroblasts. This included the “inflammatory,” “chemotactic,” “fibrogenic,” and “POSTN+” sub-populations (**Fig. 5c**). These sub-populations mainly were associated with positive regulation of inflammatory response, immune system processing, and ECM remodeling (**Fig. 5d**). Although the chemokines CXCL1, CXCL3, CXCL5, CXCL6, and CXCL8 were upregulated in UC vs. NL fibroblasts, overexpression of additional genes including IGFBP5, FTL, TGFBI, collagen, laminin, and HLA-A was observed in UC fibroblasts compared with normal. A selected number of these genes (including some of the genes mentioned in the reviewer’s comment above) are shown as volcano plots in **Fig. 5e** to highlight their upregulation in UC vs. normal fibroblasts. We included a list of differentially expressed genes in UC vs. normal fibroblasts in the supplemental information.

5c Proportion plot of the cell sub-populations in UC fibroblasts **5d** Enriched GO terms in UC vs. normal fibroblasts shown as REVIGO scatterplots ($p < 0.001$). The terms extend along the X-axis based on similarity in the type of biological process (semantic space X); difference in color highlights the variety of biological processes. Each circle represents a unique GO term. Circle size corresponds to the number of the genes associated with the GO term. **5e** Representative violin plots of the key genes responsible for the immune/inflammatory response, and ECM remodeling in UC FBs.

- 2) Can they better characterize the mesenchymal cells present in the differentiated model in the light of recent single cell analysis ie do their mesenchymal cells display niche features such as expression of Gli1/F3/Sox6 atypical Wnts/BMPs etc that are dysregulated in iHUCOs. This would better define the nature of cross-talk between fibroblasts and epithelial cells in their cultures in health vs UC that give rise to the colitis phenotype.

SnRNA-seq results revealed that iHUCO stroma recapitulated their parental signatures.

Response: Similar to UC fibroblasts, we captured the presence of unique sub-populations such as “inflammatory,” “fibrogenic,” and “POSTN+” cells in iHUCO mesenchyme, which were not identified in the iHNOs (**Fig. 6b** and **Fig. 7b**). Further, our Reactome pathway analysis on the stromal compartments of iHUC vs. iHN organoids revealed hallmarks of fibrosis including $TGF\beta$ signaling, excessive collagen production, and VEGF-mediated permeability in the stroma of the iHUCOs (**Fig. 6c**). The stromal compartment works jointly with epithelium to remodel the ECM proteins and facilitate cell migration/angiogenesis, highlighted as Netrin-1 signaling in **Fig. 6c**. Further, the Reactome terms such as “anchoring fibril formation” and “laminin interactions” with very high enrichment scores in both epithelial and stromal compartments demonstrate the vital role of epithelial/ECM

interactions in iHUCO development. Moreover, comparing the iHUCO to iHNO stroma, we identified a dysregulated immune signature, including dendritic and lymphocytes in the stroma of iHUCOs (**Fig. 6b**), as we observed in UC fibroblasts (**Fig. 5d**). A list of differentially expressed genes such as upregulated *GLI2/3* and *BMP5/7UC* in the stroma of iHUCOs compared with iHNOs are included in the supplemental information.

6b Proportion plots illustrating the unique sub-populations in iHUCO stroma **6c** Reactome pathway analysis of iHUCO stromal compartment vs. iHNO shown as bar graphs for highly enriched and significant terms in iHUCO stroma.

- 3) Can the authors provide some more characterization of the epithelial cells produced in this system. Can they detect other subtypes apart from goblets and EECs? Can they detect dysregulation of epithelial expressed defense pathways and an interferon response in iHUCOs?

Response: Although the epithelium of iHUC and iHN organoids shared some cell subtypes including cycling TA, enteroids, and tuft cells, iHUCOs showed a unique secretory signature including the presence of immature (vs. mature in iHNO) goblet cells along with presence of EECs (**Fig 6b, Supplementary Fig. 5b**). This is consistent with our *in vivo* observations confirming a lack of goblet cells and a dramatic increase in the number of CgA+ (EEC marker) cells in the crypts formed by HUCOs but not iHNOs (**Fig. 3f, g, h, i**). We also identified a unique signature in the epithelial stem cells of iHUCOs vs. iHNOs (**Fig. 7d**). Further, the Reactome pathway analysis on the epithelial compartment of iHUCOs vs. iHNOs revealed the role of IL-17 signaling with downstream regulation of NFkB and MAP kinases, along with Rho-GTPase signaling, and adherens junctions significantly enriched in iHUCO epithelium (**Fig. 6d**). These results confirmed our GO analysis of iHUCOs, highlighting the enriched terms related to adherens junctions as well as the roles of IFN- γ and TNF- α signaling noted in iHUCOs but not observed iHNOs (**Fig. 4f**).

6b Proportion plot of the sub-populations in iHUCO epithelium **7d** Volcano plot highlighting the most significant up- and down-regulated genes in stem cell sub-population of iHUCOs. **6d** Reactome pathway analysis of iHUCO vs. iHNO epithelial compartment; shown as bar graphs for highly enriched and significant terms in iHUCOs.

- 4) In the discussion it would be useful to discuss the relative advantages of their model over non-iPS derived human intestinal epithelial organoids for studying barrier function in health and IBD in the light of their findings.

We thank the reviewer for this comment and updated the first three paragraphs of the discussion accordingly:

The nature of UC is complex as it relies on interaction of different cell compartments, and this has made it challenging to study the pathogenesis of colitis. Current therapeutic approaches in IBD mainly focus on the suppression of immune responses¹; however, these therapies often fail, suggesting that the role of the epithelial and mesenchymal components in the development and progression of UC is not appreciated.

Common in vivo rodent models or epithelial organoids as an in vitro model have advantages but do not adequately recapitulate the complexity and etiology of clinical UC². In particular, epithelial organoids focus solely on the epithelium and do not include the role of microenvironment in colitis³⁻⁵.

Our induced in vitro model of human UC (iHUCO) includes both functional epithelial and stromal compartments and retains the complexity of the colitic phenotype of the tissue of origin, such as a compromised epithelial barrier, indicating a significant advance in UC modeling. Currently, in attempts to understand the mechanism by which inflammatory stroma can damage the integrity of the epithelial barrier, many studies try to develop a direct co-culture of IECs, mesenchyme, and T cells. However, several layers of challenges, for example, histocompatibility issues, result in reduced mimicry of the in vivo dynamic between various cell types. Here for the first time, we introduce a human in vitro model that will allow us to identify and understand potential mechanisms behind the direct interaction of IECs and stromal cells in a patient-specific manner. Further, our model provides a comprehensive human model to manipulate immune and inflammatory regulators, naturally present in iHUCOs, to identify potential therapies to recover functioning epithelial cells in patients suffering from UC. With such a model, the exploration of strategies for personalized medicine can be facilitated, and the mechanisms underlying the pathophysiology of human UC may be investigated in vitro.

Minor comments

Line 45 “leading to colectomy” would be better than “leading to ultimate resection of the colon and rectum”

Response: *We thank the reviewer for this detailed observation and rephrased “leading to ultimate resection of the colon and rectum” with “leading to colectomy.”*

Point-by-point Response to Reviewer #2

Reviewer 1 stated that “The work of Sarvestani et al. gives new insights into the pathogenesis of UC by aid of a patient-derived in vitro model recapitulating UC, which makes this study of major interest in the field of ulcerative colitis as there is a major interest in understanding of UC pathogenesis. The manuscript describes a novel *in vitro* model of induced organoids from patients using induced pluripotent stem cells. The manuscript is well written and set up in a logical manner. The subject of the article is very suitable for publication in *Nature Communications*.”

Response: *We are truly appreciative of the reviewer’s enthusiasm and positive feedback, which were very helpful to improve the validity of our data. Our response to the reviewer’s comments follow:*

Major comments:

- 1) I suggest to increase number of patient tissue specimens and induced organoids for studies to at least n=10.

Response: *We thank the reviewer for this comment. In our limited timeframe, in addition to those already presented in the original manuscript, we added 3 reprogrammed lines from a colitic origin and 2 from a normal colon origin, therefore bringing the total to n=11, and incorporated the new data in the relevant graphs, accordingly.*

- 2) I suggest that the quantification of IHC and IF is performed by two independent individuals.

Response: *We appreciate the reviewer’s valuable suggestion. In addition to unbiased quantification from 2 individuals in the laboratory, we asked two pathologists, Dr. Michael Cruise and Dr. Haiyan Lu, to review and validate the quantification approaches and the data.*

References

1. Neurath MF. New targets for mucosal healing and therapy in inflammatory bowel diseases. *Mucosal immunology*. 2014;7(1):6-19.
2. Goyal N, Rana A, Ahlawat A, Bijjem KR, Kumar P. Animal models of inflammatory bowel disease: a review. *Inflammopharmacology*. 2014;22(4):219-233.
3. Sato T, Stange DE, Ferrante M, et al. Long-term expansion of epithelial organoids from human colon, adenoma, adenocarcinoma, and Barrett's epithelium. *Gastroenterology*. 2011;141(5):1762-1772.
4. VanDussen KL, Marinshaw JM, Shaikh N, et al. Development of an enhanced human gastrointestinal epithelial culture system to facilitate patient-based assays. *Gut*. 2015;64(6):911-920.
5. Sarvestani SK, Signs SA, Lefebvre V, et al. Cancer-predicting transcriptomic and epigenetic signatures revealed for ulcerative colitis in patient-derived epithelial organoids. *Oncotarget*. 2018;9(47).

REVIEWERS' COMMENTS

Reviewer #1 (Remarks to the Author):

The authors have provided additional data and adequately answered the reviewers queries.

Reviewer #2 (Remarks to the Author):

In this re-submission of their manuscript describing a novel model to study Ulcerative colitis in vitro, the authors have offered response and new data to the multiple points of criticism raised in the original critique. However, the integration of these new data within the text should be reconsidered to better streamline the manuscript. These novel results are understated due to incomplete or incorrect presentation.

Additional comments

The marker plots in Supplementary Fig. 4a and Fig.7a should be labelled additional with NL/UC FBs or iHNO/iHUVO to visualize the origin of the different created clusters. Has the colour code in the corresponding UMAPs a special meaning? Is the colour code corresponding to a heat map and if yes which one?

Supplementary Fig. 5a lack of labelling. Which UMAPs represent what? In addition, numbers at the pie chart would support the finding. Furthermore, the figure legend is incorrect. Please revise your figure legends carefully.

Fig.6 and Fig.7 include corresponding results for iHUCOs and iHNOs. Please reconsider a new arrangement of the figures that readers are able to easy follow the study.

The novel experiments and figures included in the revised version of the manuscript expand and support the story in several aspects. However, these additional findings should be discussed in more detail and critically compared and differences and similarity of UC tissue, fibroblasts and iHUCOs should be highlighted. A comparison, or as suggested in our previous comments, correlations with the colitis phenotype would strengthen the manuscript.

Reviewer #1 (Remarks to the Author):

The authors have provided additional data and adequately answered the reviewers queries.

Response: *We appreciate reviewer #1's acknowledgement of our efforts.*

Reviewer #2 (Remarks to the Author):

1. In this re-submission of their manuscript describing a novel model to study Ulcerative colitis in vitro, the authors have offered response and new data to the multiple points of criticism raised in the original critique. However, the integration of these new data within the text should be reconsidered to better streamline the manuscript. These novel results are understated due to incomplete or incorrect presentation.

Response: *We thank the reviewer for appreciating our significant effort in this regard. We have significantly rearranged figures 6, 7 along with supplementary figures 4 and 5 to clarify and to allow readers to more easily follow the logic. We have also increased the labelling within the figures, and dedicated more text to the significance of our findings in the context of the field, including comparisons to multiple manuscripts which are now referenced both in the results and in the discussion.*

Additional comments

2. The marker plots in Supplementary Fig. 4a and Fig.7a should be labelled additional

with NL/UC FBs or iHNO/iHUV0 to visualize the origin of the different created clusters. Has the colour code in the corresponding UMAPs a special meaning? Is the colour code corresponding to a heat map and if yes which one?

Response: Both figures have the marker plot groups colored to match the parental fibroblasts. These colors are indicated along the x-axis such that the turquoise color is used for the normal fibroblasts and the pink color indicates the UC fibroblasts. If there are clusters in the UMAP in which both types are represented, then the cluster labelling in the x-axis of the marker plot is colored half-half turquoise and pink. The color code in the clusters in the larger UMAP does not have any significance, just used to discriminate the clusters.

3. Supplementary Fig. 5a lack of labelling. Which UMAPs represent what? In addition, numbers at the pie chart would support the finding. Furthermore, the figure legend is incorrect. Please revise your figure legends carefully.

Response: Supplementary Fig. 5a now has labelling for which UMAP is which (NL vs. UC), and the pie charts now have the percentages of cell types within the pie sector directly applied. Further, the figure legend is now revised as 5a had been switched with 5b.

4. Fig.6 and Fig.7 include corresponding results for iHUCOs and iHNOs. Please reconsider a new arrangement of the figures that readers are able to easily follow the study.

Response: Thank you for this helpful suggestion. As suggested, these figures have been rearranged to more clearly delineate, at the single nuclear level, the differences between the transcriptional signatures indicating the presence of immune cells in the iHUCOs compared to the iHNOs. The reader no longer has to flip back and forth between the figures. In parallel, the text now reflects these comparisons, first at the transcriptional level, then also at the protein levels for HLA-A and LIMCH1. Figure 7g now leads the reader back to the rationale for investigation of the CXCL8-CXCR1 axis, by comparing the cytokine profiles for both parental fibroblasts and for mesenchyme derived fibroblasts.